# Multifaceted information-seeking motives in children

Gaia Molinaro [1,2,3] ✉, Irene Cogliati Dezza [1,2,4], Sarah Katharina Bühler[1,2], Christina Moutsiana[5] & Tali Sharot [1,2,6] ✉

From an early age, children need to gather information to learn about their environment. Deciding which knowledge to pursue can be difficult because information can serve several, sometimes competing, purposes. Here, we examine the developmental trajectories of such diverse information-seeking motives. Over five experiments involving 521 children (aged 4–12), we find that school-age children integrate three key factors into their information-seeking choices: whether information reduces uncertainty, is useful in directing action, and is likely to be positive. Choices that likely reveal positive information and are useful for action emerge as early as age 4, followed by choices that reduce uncertainty (at ~age 5). Our results suggest that motives related to usefulness and uncertainty reduction become stronger with age, while the tendency to seek positive news does not show a statistically significant change throughout development. This study reveals how the relative importance of diverging, sometimes conflicting, information-seeking motives emerges throughout development.

From a very young age, infants seek information through gestures and vocalization[1–3]. Once proficient in verbal communication, children may ask up to 95 questions in a single hour to support their learning[4,5]. Because information can serve several, sometimes competing, goals, deciding which information to seek out can be a particularly difficult problem to solve. Imagine, for example, a child who suspects their sibling received more pieces of candy than they did. Counting the candy will reduce their uncertainty regarding who received more but may also trigger animosity. As a result, the child will need to arbitrate between different information-seeking motives before choosing a plan of action. Whether the child decides to count the candy may depend on how much they value uncertainty reduction versus their desire to avoid animosity and/or on whether they believe they will be able to correct the potential injustice. Here, we quantify developmental changes to the relative importance of different motives for information-seeking.

We have recently proposed a theory that characterizes three key motives for information-seeking[6]. According to this theory, when deciding whether to seek information, people first estimate what the information will reveal and then estimate the expected impact of that information on their affect (i.e., how the information will make them feel), cognition (how the information will improve their models of the world) and action (how the information will be useful for obtaining external rewards and avoiding external punishment). In particular, all else being equal, people will be more likely to seek information (1) when they expect knowledge to make them feel better[7–18], (2) when uncertainty is high[7,14,18–24], and (3) when it can aid in selecting action that will help gain rewards and avoid harm[7,17,25–27].

Different people assign different weights to each of these factors when deciding whether to seek information[7,13]. These are integrated into a calculation of the value of information, which results in individual differences in information-seeking behavior[7,13]. For example, an

[1]Affective Brain Lab, Department of Experimental Psychology, University College London, London WC1H 0AP, UK. [2]Max Planck University College London Centre for Computational Psychiatry and Ageing Research, London WC1B 5EH, UK. [3]Department of Psychology, University of California, Berkeley, 2121 Berkeley Way, Berkeley, CA 94704, USA. [4]Department of Experimental Psychology, Ghent University, 9000 Ghent, Belgium. [5]Department of Social Sciences, University of Westminster, London W1W 6UW, UK. [6]Department of Brain and Cognitive Sciences, Massachusetts Institute of Technology, Cambridge, MA, USA. ✉e-mail: gaiamolinaro@berkeley.edu; t.sharot@ucl.ac.uk

individual who puts most of the weight on how information may impact their affective state may decide to skip a medical screening, while someone who puts most of the weight on whether information is useful in avoiding harm may attend them religiously.

Studies have shown that the affective evaluation of information is associated with activity in the dopamine-rich midbrain (ventral tegmental area and substantia nigra), while the instrumental utility of information and its ability to reduce uncertainty is associated with activity in the frontal and prefrontal cortex[8,11,26]. Because subcortical regions mature earlier than frontal and prefrontal areas[28], we hypothesize that the weight assigned to affect may be fully apparent earlier in development, while that related to instrumental utility and uncertainty reduction may continue to increase throughout childhood.

While there are clues in the literature that children, like adults, prefer information that is positive[29], can guide action[30,31] and reduce uncertainty[32-37], studies have yet to disentangle the importance of these (sometimes) competing motives. Rather, motives have either been studied in isolation or in a situation where they are confounded (e.g., refs. 29,30,33,34,38). In real life, conflicting motives for knowledge are often present; thus, a complete picture of the development of information-seeking requires characterizing the developmental trajectory of multiple, coexisting information-seeking motives.

Here, we examine how the relative importance of different information-seeking motives alters with development from preschool (4-year-olds) to the beginning of teenage years (12-year-olds). We develop a child-friendly task in which, on each trial, we simultaneously vary the instrumental utility of information that children can obtain, the level of uncertainty they have about the information and the likely valence of it. By examining children's decisions of whether to seek information, we determined the age at which children begin to consider the influence of knowledge on their action, affect and cognition in such multifaceted situations. We find that as early as preschool years, children seek information that is likely to yield positive news, is useful for guiding action, and can reduce uncertainty. The tendency to seek useful and uncertainty-reducing information strengthens with age, while the drive to seek positive news does not show a statistically significant change throughout development.

## Results

In Experiment 1, 204 children (age M = 7.44 ± 2.38, age range = 4–12, 47.5% female, 52.5% male) completed an information-seeking paradigm online. We then ran a full replication in Experiment 2 (N = 196 children, age M = 7.62 ± 2.27, age range = 4–11, 48% female, 52% male).

The task was a child-friendly version of an adult task we had developed[7]. It was presented as a story about two fishermen who were sitting beside opposite ponds, each pond having six items in it—either big fish, small fish, big cans, small cans or seaweed. Participants were told that cans and fish may be hiding behind seaweed. Either a can or fish (big or small) was already attached to the fishing hook, but the participant did not know which one. The fishermen's goal was to catch fish (bigger is better) and avoid catching cans (bigger is worse; Fig. 1), and participants were asked to help. There were two actions participants could take. First, they had an opportunity to get more information on which item was already hooked. In particular, they were to press a button (called a "flashlight") on either the left or right side of the pond, which would automatically remove three of the decoy items, leaving the item that was attached to the hook as well as two decoys. In addition, any remaining seaweed was removed from the selected pond so the items behind them could be seen. Then, for each pond, either the fisherman or the participant could decide whether to lift the fishing pole and collect the item attached to it or move to the next trial without risking catching a can. Neither the fisherman's decision nor the fishing outcome was revealed.

Similar to our previous experiment in adults[7], unbeknownst to participants, we varied three factors on each trial: (1) the likelihood that the information would convey "good news"—this was varied by manipulating the proportion and size of fish and cans and thus the expected value (EV) of the outcome; (2) the amount of uncertainty regarding the outcome—varied by manipulating the number of seaweeds; (3) the instrumental utility of information—varied by manipulating the likelihood that the participant would make the pole-lifting decision versus that the fisherman would make it, which varies the agency a participant has over the outcome decision. If the participant cannot make the pole-lifting decision, then information has no instrumental utility (see ref. 7) because the participants were told the fisherman was not able to see the items in the pond. For full task details, see the "Methods" section and Fig. 1.

### Preliminary checks

To ascertain whether children of all ages correctly understood the task instructions, we included multiple questions to measure their

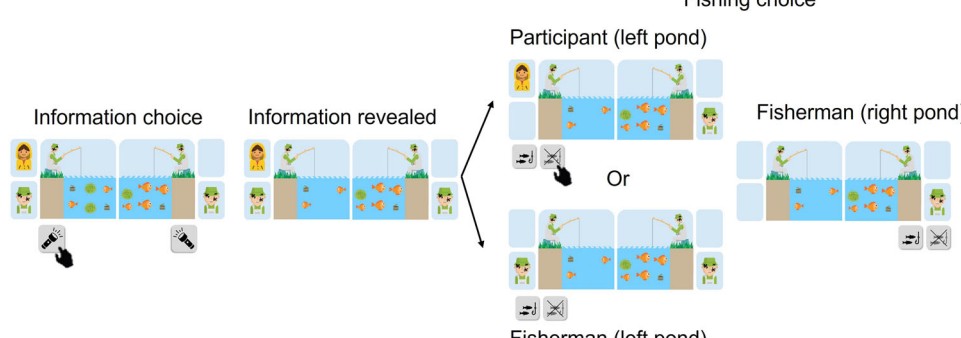

**Fig. 1 | Information-seeking task.** In each of the 30 trials, participants viewed two fishermen sitting next to two ponds. The fishermen's goal was to catch fish and avoid cans, and the participants were to help them. In each pond was a combination of fish, cans, and seaweed, and one item in each was already attached to the hook, though it was impossible to know which one. Participants indicated if they wanted more information about the items in the right or left pond (information choice). In this example, the participant decided that they wanted more information about the left pond. Thus, three decoy items that were not attached to the pole were removed from the left pond (information revealed). In addition, if any seaweed remained, the items hiding behind them were revealed. Only the participant, but not the fishermen, could see which items were left. Subsequently, either the participant or the fisherman would decide to reel in the hook on each pond (fishing choice). If only an image of a fisherman (green hat) was shown next to the pond (as can be seen in this case in the right pond), it was the fisherman who would ultimately make the decision on whether to raise the pole on that pond. If only an image of a child (yellow hoodie) was shown next to the pond, then the participant would make the decision on whether to raise the pole in that pond. If both images were shown (as can be seen in this case on the left side of the pond), either the participant or the fisherman would eventually make the decision with a 50% chance. Fisherman icon: Flaticon.com.

**Table 1 | Participants across ages obtained excellent scores on all control measures**

| Exp. | Age (years) | Instructions comprehension M ± SD | EV comprehension | | | | Correct fishing | | | |
|---|---|---|---|---|---|---|---|---|---|---|
| | | | M ± SD | Test statistic | p | r | M ± SD | Test statistic | p | r |
| 1 | 4–5 | 0.89 ± 0.10 | 0.82 ± 0.17 | V(52) = 1310 | <0.001 | 0.87 | 0.62 ± 0.15 | V(52) = 1151.5 | <0.001 | 0.63 |
| | 6–7 | 0.96 ± 0.06 | 0.90 ± 0.13 | V(53) = 1429.5 | <0.001 | 0.88 | 0.73 ± 0.14 | V(53) = 1473 | <0.001 | 0.86 |
| | 8–9 | 0.99 ± 0.02 | 0.96 ± 0.09 | V(47) = 1128 | <0.001 | 0.91 | 0.84 ± 0.11 | V(47) = 1175 | <0.001 | 0.87 |
| | 10–12 | 0.99 ± 0.03 | 0.96 ± 0.09 | V(48) = 1225 | <0.001 | 0.92 | 0.85 ± 0.14 | V(48) = 1174 | <0.001 | 0.87 |
| 2 | 4–5 | 0.91 ± 0.08 | 0.86 ± 0.14 | V(41) = 861 | <0.001 | 0.88 | 0.68 ± 0.14 | V(41) = 772 | <0.001 | 0.83 |
| | 6–7 | 0.94 ± 0.07 | 0.88 ± 0.17 | V(51) = 1221 | <0.001 | 0.89 | 0.66 ± 0.14 | V(51) = 1180 | <0.001 | 0.79 |
| | 8–9 | 0.98 ± 0.04 | 0.96 ± 0.08 | V(47) = 1176 | <0.001 | 0.91 | 0.78 ± 0.15 | V(47) = 1161.5 | <0.001 | 0.85 |
| | 10–12 | 0.98 ± 0.03 | 0.96 ± 0.11 | V(53) = 1483.5 | <0.001 | 0.94 | 0.83 ± 0.15 | V(53) = 1425 | <0.001 | 0.86 |

Scores on the instructions comprehension test, expected values (EV) comparisons task, and proportion of correct "fishing" choices were excellent for all age groups in both Experiment 1 and Experiment 2. Two-tailed Wilcoxon Signed Rank tests were performed against chance level (i.e., 0.5).

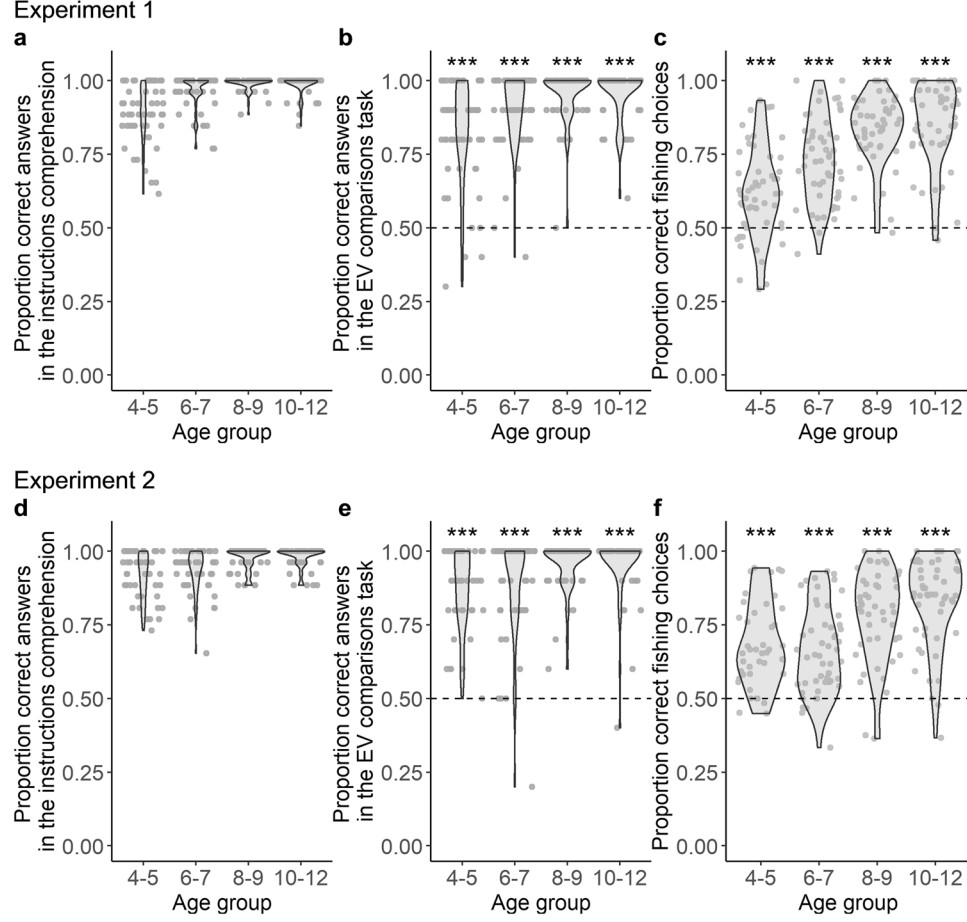

**Fig. 2 | Participants show good task comprehension and ability to compare the expected value (EV) of different options in the context of our task.** Participants in all age groups **a**, **d** showed good task comprehension, **b**, **e** were able to indicate which pond in a pair was "better"–i.e., had a larger EV, and **c**, **f** were more likely to "fish" when EV was positive than negative. The top row shows the results for Experiment 1, the bottom row for Experiment 2 (replication). *** Wilcoxon Signed Rank two-tailed test with $p < 0.001$.

comprehension of the task (see Supplementary Methods). Scores on these questions revealed excellent comprehension. To verify that children of all ages (1) understood the instructions, (2) were able to compare expected values, and (3) performed adequately in the task, we split children into the following age groups 4–5 (preschoolers), 6–7 (early elementary), 8–9 (readers), and 10–12 (older children) year-olds (based on ref. 39; Table 4). Comprehension scores (on a scale from 0 being worst and 1 being perfect) were excellent for all age groups, as observed in Table 1 and Fig. 2a, d.

After participants completed the information-seeking task, we also included additional trials that tested participants' ability to compare expected value (EV). In these trials, participants were shown two ponds with different combinations of items and had to select the better one (see "Methods"). Participants of all ages performed well above chance (on a scale from 0 being all wrong to 1 being all correct, with chance level at 0.5), as observed in Table 1 and Fig. 2b, e.

All age groups were also more likely to fish when EV was positive than when it was negative, thus showing both comprehension of the

**Table 2 | Results from a logistic mixed-effects model (Experiments 1 and 2)**

| Variable | $\beta$ | | S.E.M | | $X^2$ | | $p$ | | $\omega$ | |
|---|---|---|---|---|---|---|---|---|---|---|
| | Exp. 1 | Exp. 2 | Exp. 1 | Exp. 2 | Exp. 1 | Exp. 2 | Exp. 1 | Exp. 2 | Exp. 1 | Exp. 2 |
| ΔEV | 0.59 | 0.67 | 0.07 | 0.07 | 70.59 | 82.89 | <0.001 | <0.001 | 0.59 | 0.65 |
| Δagency | 0.9 | 0.66 | 0.08 | 0.06 | 128.71 | 122.29 | <0.001 | <0.001 | 0.79 | 0.79 |
| Δuncertainty | 0.27 | 0.31 | 0.04 | 0.05 | 41.59 | 46.44 | <0.001 | <0.001 | 0.45 | 0.49 |
| Age | 0.08 | −0.07 | 0.06 | 0.05 | 2.11 | 2.29 | 0.146 | 0.13 | 0.10 | 0.11 |
| Instructions comprehension | 0.01 | −0.02 | 0.06 | 0.05 | 0.02 | 0.19 | 0.897 | 0.661 | 0.01 | 0.03 |
| EV comprehension | −0.1 | 0.12 | 0.06 | 0.05 | 3.19 | 7.58 | 0.074 | 0.006 | 0.13 | 0.20 |
| Correct fishing | 0 | 0.07 | 0.06 | 0.05 | 0 | 2.06 | 0.956 | 0.151 | 0 | 0.10 |
| ΔEV * Age | 0.11 | 0.03 | 0.07 | 0.07 | 2.86 | 0.14 | 0.091 | 0.71 | 0.12 | 0.03 |
| Δagency * Age | 0.34 | 0.17 | 0.08 | 0.06 | 24.19 | 10.12 | <0.001 | 0.001 | 0.34 | 0.23 |
| Δuncertainty * Age | 0.2 | 0.14 | 0.04 | 0.04 | 23.85 | 10.33 | <0.001 | 0.001 | 0.34 | 0.23 |

The results reveal that children seek information more on the side of the pond where the probability of receiving good news is greater (ΔEV), agency is greater (Δagency), and uncertainty (Δuncertainty) is greater. While the importance of Δagency and Δuncertainty increased with age, as evidenced by significant interactions, that of ΔEV remained constant.

task and an intuition of EV, as observed in Table 1 and Fig. 2c, f. Taken together, our checks reveal that participants understand the task well and can intuitively assess the expected value of item combinations. Having verified these aspects, we proceed to analyze developmental patterns in information-seeking.

## Key independent variables

Based on our theory of information-seeking motives[6], we expected that participants would want information more when (1) the probability of receiving good news was greater, (2) uncertainty was greater and (3) the instrumental utility of information was greater. We theorized that these three factors would be integrated to guide information-seeking. To test this prediction, we varied the following factors in each of the two available information-seeking options: (1) the expected value (EV) of the outcome (i.e., the average value of cans, fish and seaweed in a pond, ranging from +20 if all items were big fish to −20 if all items were big cans; higher EVs correspond to greater probability of receiving good news); (2) the amount of uncertainty about the possible outcomes by varying the amount of seaweed in the pond (ranging from zero to six out of six items); and (3) the probability that the participant, rather than the fisherman, would be given the opportunity to make the decision of whether to raise the pole to fish (either 0, 0.5, or 1). Importantly, as only participants, but not fishermen, were able to see inside the pond, information on which items were likely hooked to the pole had instrumental utility only when the participant had agency to make the decision on whether to raise the pole. In each trial and pond, uncertainty over the outcome (i.e., which item was attached to the hook) could be driven not only by the amount of seaweed in the pond but also by the standard deviation of the items in the pond (e.g., ref. 7)—that is, the standard deviation from the mean value of all visible items in the pond. Thus, we made sure to construct the trials such that the ponds with a greater amount of seaweed also had greater standard deviation (see "Methods").

We constructed trials such that the difference between the right and left ponds (i.e., Δ) for each of the three factors described above was orthogonal. That is, the Δs of the three factors did not correlate across trials (see "Methods" for pairwise Pearson's correlation coefficients). This should maximize decision conflict and allow us to estimate the effect of each factor on information-seeking.

In all the analyses below, the key independent variables of interest were ΔEV (EV right pond − EV left pond), Δagency (agency probability in the right pond − agency probability in the left pond), and Δuncertainty (number of seaweeds in the right pond − number of seaweeds in the left pond).

## Developmental trajectories for information-seeking motives

To characterize the developmental trajectories of the weight assigned to each information-seeking factor, we first ran a logistic mixed-effects model predicting children's information-seeking choices (coded as 0 if the participant selected the left pond and 1 for the right pond) from the following factors (as fixed effects and random slopes for each participant): ΔEV, Δagency, and Δuncertainty. Age (in years) was included as a fixed effect, as was its interaction with each of the above Δs. We also controlled for the following fixed effects: participants' scores in the instruction comprehension test, participants' scores on the EV comparison task, and the proportion of correct "fishing choices" each participant made.

As observed in Table 2 and Fig. 3, the results of the mixed models reveal that all three motives were significantly related to children's information-seeking choices. Children were more likely to seek information for the pond where (1) the likelihood of receiving good news was higher (i.e., higher EV); (2) uncertainty was greater and (3) the likelihood of making the fishing choice was higher (i.e., higher agency). Note that these effects were also found in adults performing the same task (Experiment 6 and Experiment 7 below). Additionally, the weight assigned to uncertainty and agency during information-seeking increased with age, as observed by significant interactions, while the weight assigned to EV did not interact with age. The same results were observed when running a model that includes triple interactions between each pair of information-seeking motives (ΔEV, Δagency, Δuncertainty) and age (see Supplementary Table 2 for statistics). This model also revealed that none of the interactions were significant (Supplementary Table 2). The same results were also observed when restricting the analysis to trials where Δagency was zero, i.e., information-seeking choices could not be explained by instrumental utility (see Supplementary Table 3 for statistics).

To examine the robustness of these results, we performed a complementary analysis. In particular, for each participant, we ran a regularized (ridge) logistic regression predicting information choice from ΔEV, Δuncertainty and Δagency. We then correlated participants' $\beta$ coefficients of each of the three factors separately with their age in years. We found that, as age increased, participants assigned more weight to Δuncertainty (Experiment 1: $r(202) = 0.28$, $p < 0.001$, 95% CI = [0.15, 0.40]; Experiment 2: $r(193) = 0.19$, $p = 0.007$, 95% CI = [0.05, 0.32]) and Δagency (Experiment 1: $r(202) = 0.32$, $p < 0.001$, 95% CI = [0.18, 0.43]; Experiment 2: $r(193) = 0.21$, $p = 0.003$, 95% CI = [0.07, 0.34]) when seeking information. In contrast, the correlation between age and the weight participants' assigned to ΔEV when seeking information was not significant (Experiment 1: $r(202) = 0.09$, $p = 0.200$, 95%

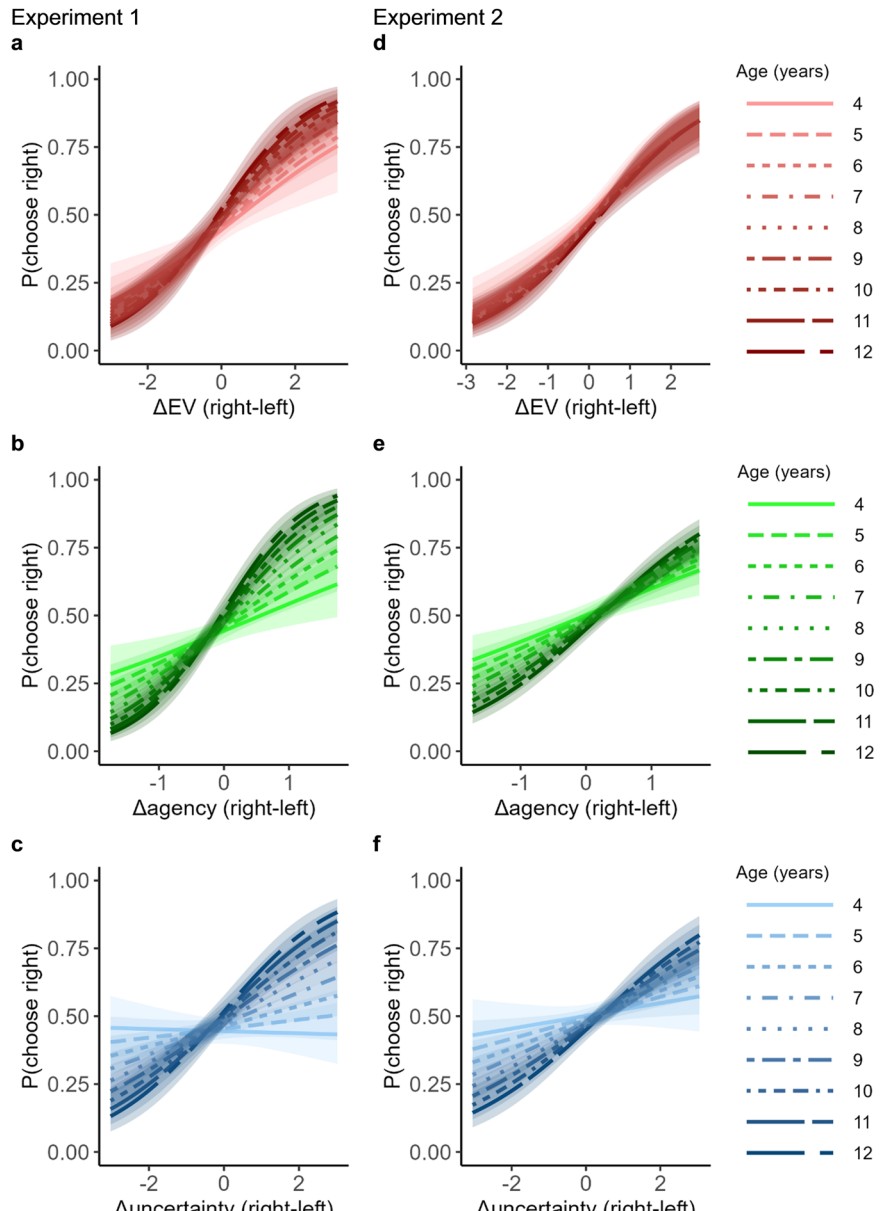

**Fig. 3 | Different motives for information-seeking follow different developmental trajectories.** Each line reflects the estimated probability of choosing to seek information on the right based solely on either ΔEV (**a**, **d**), Δagency (**b**, **e**), or Δuncertainty (**c**, **f**) when all other factors are held constant (using the logistic mixed-effects model described above) for a specific age. **a**, **d** The opportunity to receive good news (measured via expected value) was associated with information-seeking similarly across ages (as observed by the smaller divergence of lines), while the importance of **b**, **e**, the instrumental utility of information (varied by manipulating agency) and **c**, **f** the relative importance of uncertainty on information-seeking, increase as children get older (as observed by the wide divergence between the lines in **b**, **e**). The left side shows results for Experiment 1, the right side for Experiment 2. Shading represents 95% CIs.

CI = [−0.05, 0.22]; Experiment 2: $r(193) = 0.01$, $p = 0.856$, 95% CI = [−0.13, 0.15]). Bayes factors for the correlation between ΔEV $\beta$ coefficients and age were smaller than 1, signifying anecdotal evidence in favor of the null hypothesis in Experiment 1 ($BF = 0.36$) and moderate evidence in favor of the null hypothesis in Experiment 2 ($BF = 0.17$) that ΔEV did not alter with age. These results support the conclusion that preference for information that is likely to convey good news remains relatively stable throughout development, while the preference for information that can reduce uncertainty and has instrumental utility increases with development.

To estimate the age at which the effects of ΔEV, Δagency, and Δuncertainty became statistically significant, we used the Johnson-Neyman technique. In particular, this analysis uses the mixed model described above to estimate the fixed effects of ΔEV, Δagency, and Δuncertainty on information-seeking choices as a function of age and then estimates the age at which the effect becomes significant[40] (Fig. 4). The analysis revealed that the effects of good news (ΔEV) on information-seeking were already significant at age 4 (Experiment 1: $\beta = 0.43 \pm 0.12$, $p < 0.001$, 95% CI = [0.18, 0.67]; Experiment 2: $\beta = 0.63 \pm 0.14$, $p < 0.001$, 95% CI = [0.36, 0.90]; Fig. 4a, d). The tendency to select information that is useful in guiding action was also significant from the youngest tested age (Experiment 1: $\beta = 0.40 \pm 0.13$, $p = 0.002$, 95% CI = [0.15, 0.65]; Experiment 2: $\beta = 0.40 \pm 0.13$, $p < 0.001$, 95% CI = [0.19, 0.60]; Fig. 4b, e) and the tendency to select information when uncertainty was high emerged at around age 5 (Experiment 1: age 5.52, $\beta = 0.11 \pm 0.05$, $p = 0.037$, 95% CI = [0.01, 0.21]; Experiment 2: age 4.85, $\beta = 0.15 \pm 0.07$, $p = 0.037$, 95% CI = [0.01, 0.28]; Fig. 4c, f).

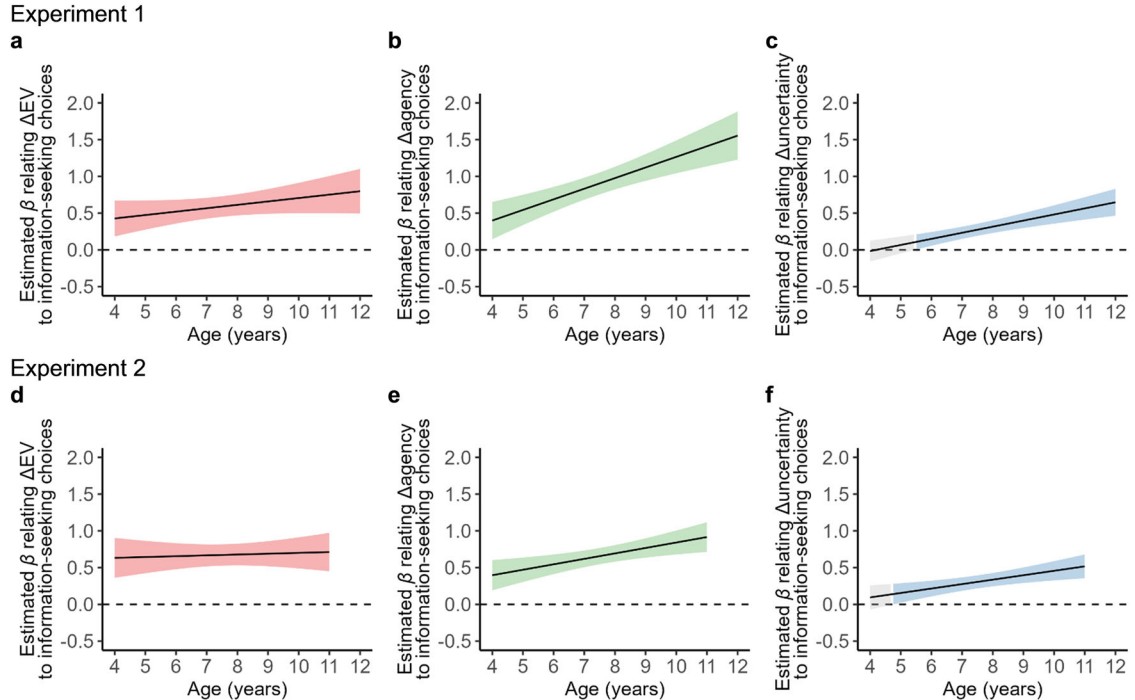

Fig. 4 | **The emergence of different motives for information-seeking across development.** In the Johnson-Neyman plots above, lines reflect the estimated fixed effects of ΔEV, Δagency, and Δuncertainty on information-seeking choices as a function of age for Experiment 1 (**a**–**c**) and Experiment 2 (**d**–**f**). Colored shading indicates the areas of statistical significance ($p < 0.05$), while gray shading indicates regions where the effects were non-significant. **a**, **d** The opportunity to receive good news (measured via EV) in information-seeking arises as early as age 4, as does the importance of **b**, **e**, the instrumental utility of information (varied by manipulating agency), while **c**, **f** the relative importance of uncertainty arises around age 5.

Individual estimates for the effects of Δagency on information-seeking correlated with the proportion of correct fishing choices—defined as the proportion of times the participant chose to fish when the EV of the pond was positive and not to fish when the EV of the pond was negative (Experiment 1: $r(202) = 0.38$, $p < 0.001$, 95% CI = [0.26, 0.49]; Experiment 2: $r(193) = 0.27$, $p = 0.001$, 95% CI = [0.14, 0.40]).

## Accounting for a simple choice strategy

We tested whether younger children were more likely to repeat the information-seeking choice they made in the previous trial. To assess this, we ran a mixed-effects logistic model predicting information-seeking as previously described, with one additional variable—the choice made in the previous trial. The results were mixed. While in Experiment 1 there was an interaction between age and previous choice ($\chi^2(1) = 9.51$, $p = 0.002$, $\omega = 0.22$), which was characterized by younger children using this strategy more, the effect was not replicated in Experiment 2 ($\chi^2(1) = 2.06$, $p = 0.151$, $\omega = 0.10$). Importantly, even after accounting for this strategy in the model, we still observed in both experiments an interaction between age and Δuncertainty (Experiment 1: $\chi^2(1) = 23.96$, $p < 0.001$, $\omega = 0.34$; Experiment 2: $\chi^2(1) = 9.27$, $p = 0.002$, $\omega = 0.21$) and between age and Δagency (Experiment 1: $\chi^2(1) = 23.92$, $p < 0.001$, $\omega = 0.34$; Experiment 2: $\chi^2(1) = 9.03$, $p = 0.003$, $\omega = 0.22$), and no interaction between age and ΔEV (Experiment 1: $\chi^2(1) = 3.01$, $p = 0.083$, $\omega = 0.12$; Experiment 2: $\chi^2(1) = 0.16$, $p = 0.686$, $\omega = 0.03$; see Supplementary Table 1 for all statistics). Thus, this simple strategy does not seem to account for developmental differences in the tuning of information-seeking to uncertainty and agency.

Moreover, even though young children were more likely to use a simple strategy in Experiment 1, in both Experiment 1 and Experiment 2 they took more time in making their information-seeking decision. In particular, relating age to average log reaction times for information-seeking choices revealed that younger children engaged with the task for longer than older children (Experiment 1:

$r(202) = -0.58$, $p < 0.001$, 95% CI = [−0.66, −0.47]; Experiment 2: $r(194) = -0.57$, $p < 0.001$, 95% CI = [−0.66, −0.47]). This suggests that younger children were indeed taking time, presumably to engage with the task. (The same was also true for raw reaction times (Experiment 1: $r(202) = -0.50$, $p < 0.001$, 95% CI = [−0.60, −0.40]; Experiment 2: $r(194) = -0.42$, $p < 0.001$, 95% CI = [−0.54, −0.31])).

## Additional control experiments and analyses

We ran three additional control experiments. Briefly, in Experiment 3, we show that the effect of agency on information-seeking is not due to children simply being drawn to their icon but rather is explained by children wanting information when it was likely to be more useful. Experiment 4 rules out the idea that significant effects of uncertainty are simply due to children being drawn to seaweed. In Experiment 5, we show that children of all ages track the differences in uncertainty between ponds and that accounting for age-specific abilities in tracking uncertainty does not alter the results. Additionally, we ran two experiments testing adults (Experiment 6 and Experiment 7) that show that the same variables that guide children's behavior also guide adults' information-seeking.

**Experiment 3 (control experiment)—understanding agency.** The aim of Experiment 3 was to test whether children ($N = 27$, age: $M = 6.93$ years ± 2.6 years, age range = 4–12 years, 59.3% female, 40.7% male) were more likely to select information on the pond where they had higher agency simply because they were drawn to the picture that represented them in the task. To that end, we asked participants to select information on either pond where they had 100% agency (i.e., only the icon of the participant was shown) or 50% agency (i.e., the icon of the participant was shown together with the icon of the fisherman). Given that the participant's icon was shown on both sides (see an example trial in Supplementary Fig. 2), being drawn to the icon could not guide information-seeking choices in this task.

Children showed a high level of comprehension of the instructions ($M = 0.94 \pm 0.07$). Importantly, as in the main experiment, $\Delta$agency predicted children's information-seeking choices ($\beta = 0.45$, S.E.M. = 0.14, $\chi^2(1) = 8.89$, $p = 0.003$, $\omega = 0.56$). These results suggest that our findings cannot be attributed simply to children being drawn to the participants' icon.

**Experiment 4 (control experiment)—understanding uncertainty (part I).** The aim of Experiment 4 was to test whether children ($N = 40$, age: $M = 7.45$ years $\pm 2.57$ years, age range = 4–12 years, 42.5% female, 57.5% male) preferred information in ponds with more seaweed because they wanted to reduce uncertainty or because they were drawn to seaweed. To distinguish these two possibilities, we introduced a new type of seaweed—seaweed with holes. The holes meant that the children could see that nothing was hiding behind the seaweed (Supplementary Fig. 3). Thus, more seaweed with holes did not increase uncertainty. If children simply liked more information in ponds with seaweed, regardless of uncertainty, they should still select to receive information in the pond with more seaweed with holes. If, however, they cared about uncertainty reduction, the amount of seaweed with holes should not drive their information-seeking decisions.

Children showed a high level of comprehension of the instructions ($M = 0.97 \pm 0.04$). Confirming our findings from the main experiment, $\Delta$uncertainty predicted children's information-seeking choices ($\beta = 0.87$, S.E.M. = 0.29, $\chi^2(1) = 10.52$, $p = 0.001$, $\omega = 0.51$). Thus, our finding that uncertainty guides children's choices cannot be attributed to children being drawn to seaweed.

**Experiment 5 (control experiment)—understanding uncertainty (part II).** The aim of Experiment 5 was to test whether children ($N = 54$, age $M = 7.24$ years $\pm 2.49$ years, age range = 4–12 years, 51.8% female, 48.2% male) were able to track uncertainty within the context of our task. To this end, we conducted a control experiment where children were asked to guess whether each fisherman was more likely to hock a fish or a can and then report their confidence in their guess (Supplementary Fig. 4). The ability to track $\Delta$uncertainty would be observed if children indicate lower confidence in the likely outcome in the pond with more seaweed relevant to the one with less seaweed. That is, a negative relationship should be observed between $\Delta$uncertainty and $\Delta$confidence (i.e., the difference in confidence between the two sides).

Uncertainty was defined (as in the main task) as the amount of seaweed. We randomized the side on which the pond with higher uncertainty was displayed. Differences in uncertainty between the two sides covered the same range that participants viewed in the main task. Participants completed an instructions comprehension task before proceeding to the main part of the experiment (see Supplementary Methods for all questions asked in this experiment).

Children showed a high level of comprehension of the instructions (4-year-olds: $M = 0.93 \pm 0.11$; 5-year-olds: $M = 0.93 \pm 0.1$; 6-year-olds: $M = 0.92 \pm 0.11$; 7-year-olds: $M = 0.98 \pm 0.04$; 8-year-olds: $M = 0.97 \pm 0.08$; 9-year-olds: $M = 0.93 \pm 0.1$; 10-year-olds: $M = 1 \pm 0$; 11-year-olds: $M = 1 \pm 0$; 12-year-olds: $M = 1 \pm 0$). Overall, $\Delta$uncertainty was a strong predictor of $\Delta$confidence ($\beta = -0.44$, S.E.M. = 0.05, $\chi^2(1) = 48.37$, $p < 0.001$), suggesting that children were able to track uncertainty in our task. A Johnson-Neyman analysis performed on the whole data set revealed that the effect of uncertainty on confidence was already significant at age 4 ($\beta = 0.17 \pm 0.08$, $p = 0.04$). As there was also an interaction between $\Delta$uncertainty and age ($\beta = -0.21$, S.E.M. = 0.05, $\chi^2(1) = 14.73$, $p < 0.001$, $\omega = 0.52$), we next accounted for these age-related differences in our main model of information-seeking. To that end, we first calculated, for each age, the Pearson correlation coefficient between $\Delta$uncertainty and $\Delta$confidence across participants. Then we entered these estimates as a control variable in the main experiment analysis by matching children to the correlation between $\Delta$uncertainty and $\Delta$confidence for the corresponding age. Even after

including this variable, we found significant effects for the interaction between $\Delta$uncertainty and age (Experiment 1: $\beta = 0.20$, S.E.M. = 0.04, $\chi^2(1) = 23.87$, $p < 0.001$, $\omega = 0.66$; Experiment 2: $\beta = 0.14$, S.E.M. = 0.04, $\chi^2(1) = 10.31$, $p < 0.001$, $\omega = 0.44$), suggesting that age-related differences in sensitivity to uncertainty during information-seeking cannot be explained by differences in the ability to track uncertainty. All other main effects of interest held true.

**Experiments 6 and 7—adults.** The aim of Experiment 6 ($N = 28$, age $M = 28.6 \pm 11.50$ years, age range = 18–66, 32.1% female, 64.3% male, 3.6% other) was to test adults' behavior on this task to examine if their information-seeking choices are also explained by $\Delta$EV, $\Delta$agency, and $\Delta$uncertainty. Experiment 7 ($N = 29$, age $M = 40.2 \pm 11.70$ years, age range = 22–64 years, 34.5 % female, 65.5% male) was a replication of Experiment 6. Adult participants completed the same task as described for Experiment 1 and Experiment 2.

Comprehension scores were excellent (Experiment 6: $M = 0.99 \pm 0.25$, Experiment 7: $M = 0.99 \pm 0.02$). Participants performed well above chance in the EV comparisons task (Wilcoxon Signed Rank two-tailed tests with chance level at 0.5: Experiment 6: $M = 0.99 \pm 0.03$, $V(27) = 406$, $p < 0.001$, $r = 0.96$; Experiment 7: $M = 0.98 \pm 0.05$, $V(28) = 435$, $p < 0.001$, $r = 0.94$). All participants were more likely to fish when EV was positive than when it was negative (Experiment 6: $M = 0.91 \pm 0.163$, $V(27) = 406$, $p < 0.001$, $r = 0.88$; Experiment 7: $M = 0.93 \pm 0.07$, $V(29) = 435$, $p < 0.001$, $r = 0.88$).

Information-seeking was positively associated with $\Delta$EV (Experiment 6: $\chi^2(1) = 12.49$, $p < 0.001$, $\omega = 0.67$; Experiment 7: $\chi^2(1) = 13.60$, $p < 0.001$, $\omega = 0.68$), $\Delta$agency (Experiment 6: $\chi^2(1) = 34.62$, $p < 0.001$, $\omega = 1.11$; Experiment 7: $\chi^2(1) = 31.94$, $p < 0.001$, $\omega = 1.05$), and $\Delta$uncertainty (Experiment 6: $\chi^2(1) = 21.71$, $p < 0.001$, $\omega = 0.88$; Experiment 7: $\chi^2(1) = 20.24$, $p < 0.001$, $\omega = 0.84$; the full model's results are reported in Table 3). These results show adults use the same three variables as children to guide information-seeking choices in this task.

## Discussion

Our findings suggest that the three-factor theory of information-seeking[6], which proposes that people consider and integrate features of information related to affect, cognition, and action when deciding what they want to know[7,13], is relevant to children as young as 5 years old. We show that when information can serve multiple purposes, preschoolers' information-seeking is guided by the ability of information to facilitate both explicit external goals and affective states (the opportunity to receive good news), and school-aged children also seek information more when it can reduce the most uncertainty. Thus, like adults (see Experiments 6 and 7), children use information-seeking to improve both internal and external outcomes (see refs. 7,13). Importantly, the data reveals that these three information-seeking motives follow different development trajectories. While most of our results suggest the tendency to seek positive information was relatively stable from preschool to the pre-teenage years, the tendency to seek information that is useful in guiding actions and when uncertainty was high increased throughout development.

It is interesting that, in children younger than about age 5, information-seeking was predominantly guided by instrumental utility and affect-related considerations. These results could not be explained by a failure to comprehend instructions or an inability to perceive expected value or uncertainty, as preschoolers showed very good comprehension of these aspects, and the one variable that requires an intuition of expected value was already significantly impacting information choices at this age. The findings do not suggest that uncertainty reduction is not important in driving information-seeking at a very young age. Rather, the implication is that in complex situations where multiple information-seeking motives compete, the relative importance of uncertainty in guiding information-seeking may be low in this young population (for a related discussion, see ref. 41). In other

**Table 3 | Adult information-seeking motives**

| Variable | $\beta$ | | S.E.M. | | $X^2$ | | $p$ | | $\omega$ | |
|---|---|---|---|---|---|---|---|---|---|---|
| | Exp. 6 | Exp. 7 | Exp. 6 | Exp. 7 | Exp. 6 | Exp. 7 | Exp. 6 | Exp. 7 | Exp. 6 | Exp. 7 |
| ΔEV | 0.68 | 1.01 | 0.18 | 0.27 | 12.49 | 13.60 | <0.001 | <0.001 | 0.67 | 0.68 |
| Δagency | 1.44 | 1.41 | 0.19 | 0.20 | 34.62 | 31.94 | <0.001 | <0.001 | 1.11 | 1.05 |
| Δuncertainty | 0.65 | 0.87 | 0.12 | 0.16 | 21.71 | 20.24 | <0.001 | <0.001 | 0.88 | 0.84 |
| Instructions comprehension | 0.13 | 0.01 | 0.13 | 0.10 | 1.05 | 0.01 | 0.306 | 0.936 | 0.19 | 0.01 |
| EV comprehension | −0.14 | −0.02 | 0.10 | 0.10 | 2.25 | 0.01 | 0.133 | 0.907 | 0.28 | 0.02 |
| Correct fishing | −0.03 | −0.04 | 0.13 | 0.09 | 0.06 | 0.30 | 0.804 | 0.586 | 0.05 | 0.10 |

Results from a logistic mixed-effect model show that expected value (EV), agency, and uncertainty guide information-seeking in adults in Experiments 6 and 7. The model was similar to the one described in the main text, but age was not included as a factor. Therefore, the model had random intercepts for each participant, fixed effects and random slopes for ΔEV, Δagency, and Δuncertainty, and fixed effects for instructions comprehension scores, EV comprehension scores, and proportion of correct fishing choices.

tasks, where motives do not compete, the influence of uncertainty is observed even in younger children (e.g., refs. 32,42.). Our results could be related to the finding that young children tend to underestimate uncertainty[43]; thus, they may not be driven by the same objective uncertainty as older children, as their subjective uncertainty differs.

All our results were replicated across two studies and were tested on a large sample of children. However, even with a large sample and replication, very small effects of age may go undetected. For example, while we found no significant effect of age on the drive to seek positive information in any of the experiments, a Bayes test revealed the null effect in Experiment 1 was anecdotal. Thus, a very small effect may exist.

Here, we were able to test diverging motives for information-seeking concurrently and yet orthogonalize these factors to tease apart their impact on information-seeking. However, future studies are required to test whether the results in children generalize to different information domains and tasks. Indeed, a limitation of this study is that we utilized a single type of task, and the exact developmental stage at which information-seeking motives emerge may be somewhat task-specific. Moreover, future studies may use our task in adolescent populations, as the development of information-seeking motives may not follow linear trajectories from childhood to adulthood. Together with neuroimaging studies, these future endeavors could shed light on the exact mechanisms underlying developmental changes in information-seeking preferences. The affective value of information has been found to correlate with activity in dopaminergic midbrain regions, while instrumental and cognitive (i.e., uncertainty-reducing) utilities of information may be represented in the frontal and prefrontal cortex (e.g., refs. 8,11,26.). As the former develops earlier than the latter regions, it is plausible that the behavioral developmental patterns we observed here align with the development of corresponding brain regions and their connectivity[28].

Children's information-seeking has traditionally been examined in terms of novelty-seeking, learning progress, and the amount of information already acquired[23,44–50]. These drives may contribute to motives examined here. For example, both novelty and learning progress motives are often related to "instrumental utility", defined as the ability of information to guide choices that will lead to external rewards. They can also contribute to what we refer to as "cognitive utility"[6], defined as the ability of information to increase an agent's understanding of the world in which they operate. Here, we test specific aspects of cognitive utility and instrumental utility—namely, uncertainty reduction and agency.

The results may have potential future applications in the fields of public policy, therapy, and education, where information dissemination needs to be targeted appropriately for different age groups (e.g., ref. 51), although these remain to be tested. For instance, our findings suggest that even preschoolers will show increased curiosity if information is presented to them as useful for their own goals or likely to yield positive news. Moreover, the model suggests that a low information value from the perspective of one motive may be compensated by a high information value with respect to other motives. For instance, if negative (e.g., sad) information needs to be conveyed, student engagement may be increased by presenting it in a way that shows how the information can be useful to achieve goals. We thus believe educators working with children (teachers, coaches, etc.) may find this knowledge useful and use it to guide communication with young students and mentees. Our research may also inspire technological advances that could provide pedagogical tools for children based on their information-seeking preferences, such as artificial learning companions[52,53]. To our knowledge, no such artificial systems currently consider hedonic, cognitive, and instrumental goals concurrently.

In sum, the results suggest that children progressively direct their learning by attuning to different needs as they grow older. The tendency to expose oneself to more positive information is relatively stable across development, while the opportunity to use information to attain external outcomes and reduce uncertainty becomes stronger with age.

## Methods
Methods were identical for Experiment 1 (the initial experiment) and Experiment 2 (the replication). Therefore, we report them together below.

### Experiment 1 and Experiment 2
**Participants**. Two hundred and nine children took part in Experiment 1, and 200 children took part in Experiment 2. Experiments were built using Gorilla Experiment Builder (https://gorilla.sc/). All children completed the experiments on Lookit (https://lookit.mit.edu/), an online platform for developmental research. Parents may have assisted children with online navigation but were asked not to interfere with the children's decisions. All participants received a fixed $10 Amazon voucher for taking part in the experiment. No additional performance bonuses were delivered. In Experiment 1, data from one child were lost due to technical issues with the online hosts. Four children failed one of the attention checks and thus their data were not analyzed. In Experiment 2, four children failed one of the attention checks and were excluded from further analyses. All other participants passed sound-checks, catch trials and instructions comprehension tests. Thus, data from 204 children were analyzed in Experiment 1, and data from 196 children were analyzed in Experiment 2 (see Table 4 for participants' demographics—split by age groups for illustration purposes). $N$ was determined based on a power analysis conducted on pilot data to achieve power of 80% with $\alpha = 0.05$ using the G*Power 3 software[54]. A sensitivity analysis revealed that our final sample sizes can detect an effect size of Cohen's $\omega = 0.20$ on a $\chi^2$-test with the desired power and one degree of freedom[55]. This suggests that the study was sufficiently powered to detect small to medium effects. However, very small yet significant effects may still go undetected.

**Table 4 | Children participants' age and sex by age group in Experiment 1 and Experiment 2**

| Age (years) | M age ± SD (years) | | % Female | | N | |
|---|---|---|---|---|---|---|
| | Exp. 1 | Exp. 2 | Exp. 1 | Exp. 2 | Exp. 1 | Exp. 2 |
| 4–5 | 4.49 ± 0.51 | 4.45 ± 0.50 | 45.3% | 59.5% | 53 | 42 |
| 6–7 | 6.44 ± 0.50 | 6.44 ± 0.50 | 50% | 57.7% | 54 | 52 |
| 8–9 | 8.54 ± 0.50 | 8.48 ± 0.50 | 47.9% | 43.8% | 48 | 48 |
| 10–12 | 10.7 ± 0.81 | 10.4 ± 0.50 | 46.9% | 33.3% | 49 | 54 |
| Overall | 7.44 ± 2.38 | 7.62 ± 2.27 | 47.5% | 48% | 204 | 196 |

All children's sex was reported as either female or male (i.e., "other" was not selected by any).

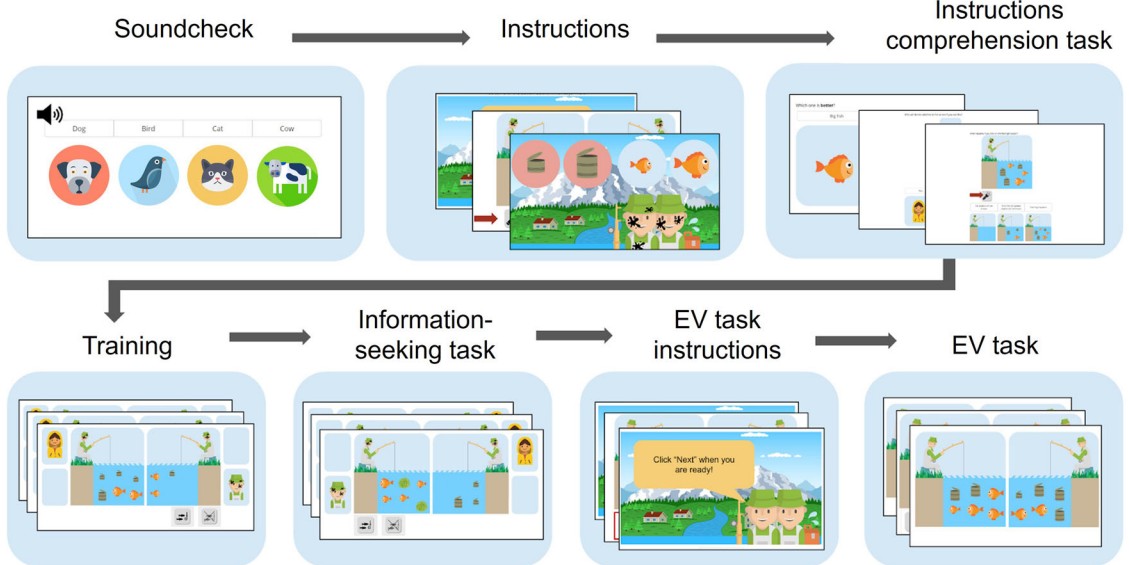

**Fig. 5 | Experimental procedure.** After providing consent, participants underwent a soundcheck to ensure the correct functioning of their computer's audio system. Next, participants read and/or listened to the task instructions and completed a comprehension test. After a brief training, participants completed the information-seeking task (see below). Finally, participants read and/or listened to the instructions of a short additional task, where they had to select the "better" pond (i.e., the one with higher expected value) and then completed that task, which is described below. Soundcheck animal images and fisherman icon: Flaticon.com.

Of note, 0.6% of children had at least one guardian with less than a high school diploma; 8.8% of children had at least one guardian with no more than a high school degree; 15.9% had at least one guardian with at most a 2-year degree; 42.2% had at least one guardian with at most a 4-year degree; 32% had both guardians (or a single guardian) with either a professional degree (such as a nursing degree, a technician degree, etc.) and/or a graduate degree. Legal guardians provided consent for their dependent minors, who then gave their assent. All participants were compensated for their time (approximately 35 min per session). The Massachusetts Institute of Technology Committee on the Use of Humans as Experimental Subjects provided ethical approval for the recruitment of children in this study. Peer Lookit users and the Lookit team also approved the study for data collection on the Lookit website. Sex information ("Female", "Male", or "Other") was provided by parents. Because we did not find significant effects of sex on our measures of interest, we analyzed data from all participants jointly. The full procedure is illustrated in Fig. 5.

**Information-seeking task.** Participants completed three practice trials before proceeding to the main task. An auditory recording guided children throughout the experiment.

The main task was set around a story of two temporarily blinded fishermen fishing in two separate ponds—presented to participants simultaneously on the left and right sides of the screen. Each fisherman wanted to catch fish while avoiding cans, and it was the participants' job to help them (Fig. 1). Within each pond, the participant would see varying combinations of six items that were either fish, cans, or seaweed. The fish and cans came in two sizes, either big or small. Participants were informed that big fish were twice as valuable as small fish and big cans were twice as harmful as small cans. Seaweed was described as covering either a fish or a can of either size. The amount and size of fish, cans, and seaweed in the water varied on each trial. Participants were told that one of the presented items was already attached to the hook, but the participants had no way of knowing which one. After observing the ponds (4 s), participants were prompted with choosing to gather further information about either the left or the right pond (self-paced). Once they selected either of the ponds, three items that were not attached to the hook would disappear from that pond, while the other pond stayed the same. Additionally, any seaweed in the selected pond would disappear to reveal the item they were hiding (4 s). Since the fishermen had trouble seeing, this information was only available to the participant. Next, either the participant (self-paced) or the fisherman (4 s) would decide whether to reel in the hook. Neither the fisherman's decision nor the fishing outcome was revealed. Whether the participant or the fisherman made this decision was dependent, for each pond, on a certain agency probability that was known to the participant: at the beginning of each trial, a set of icons was used to inform participants of the likelihood (0%, 50%, or 100%) they would choose whether to reel in the hook. The assignment of these icons to each pond and trial was random.

The likelihood that the information would convey "good news" was defined as the expected value (EV) of the outcome (which we

coded by attributing a value of 20 to large fish, 10 to small fish, −20 to large cans, −10 to small cans, and 0 to seaweed). The amount of uncertainty in each trial was varied by changing the number of seaweeds (from 0 to 6 out of 6 items). The instrumental utility of information was manipulated by varying the likelihood that the participant would decide whether to reel in the hook (i.e., the agency probability).

The combination of items presented in each pond was assigned semi-randomly on each trial. First, we randomly generated ponds with different combinations of items. We then excluded ponds in which (1) the number of seaweeds was high *and* the standard deviation of items was low and (2) ponds in which the number of seaweed was low *and* the standard deviation of items was high. This was done because we defined uncertainty as a high number of seaweeds. However, standard deviation also influences uncertainty (see, e.g., ref. 56), and we wanted to make sure that any pond in which uncertainty is high due to a large number of seaweeds was also high in terms of standard deviation. High/low was defined as greater/lower than the mean of all generated ponds. Thirty trials were semi-randomly generated for each participant, as detailed above. The correlation between standard deviation and the number of seaweeds across trials was significant (Experiment 1: $r(6118) = 0.28$, $p < 0.001$, 95% CI = [0.26, 0.30], Experiment 2: $r(5878) = 0.28$, $p < 0.001$, 95% CI = [0.25, 0.30]).

Our task was piloted on 4- and 6-year-olds, who followed the game easily and enjoyed playing. While the task may seem complex on paper (partially because we need to give the reader information that the player does not require), once played with the verbal story instructions, it seems similar to the popular digital games children play online. We performed a series of validation checks to ensure that children engaged with the task appropriately and understood each of its relevant components.

Note that following all previously published similar tasks in adults[7,13], the outcome (fish/can) on each trial was not revealed to the participant. This is crucial to model real-world situations where people seek information that provides clues about future outcomes (e.g., What gift am I likely to get for my birthday? How likely am I to get cancer?), but where those outcomes will only be known in the far future (and sometimes never). This is done because if the outcomes were revealed immediately, there would likely be a strong instrumental utility for information but little non-instrumental utility for advanced information in the form of reduced uncertainty and positive affect (for detailed explanations, see, for example, ref. 7).

### Task validation steps.

1. Sound check. After providing consent, participants completed a soundcheck wherein they were presented with a simple animal sound and asked to choose the corresponding animal from four illustrated options. All participants passed this check.
2. Comprehension check. After reading and/or listening to instructions (available in the Supplementary Methods), participants completed multiple-choice questions that tested their comprehension (see Supplementary Methods for the complete comprehension check). Participants were given up to three opportunities to answer each question correctly. Participants' responses were coded as 2 when answered correctly at the first attempt, 1 when answered correctly after one attempt, and 0 when answered correctly after two attempts. No one answered incorrectly more than twice for a single question. The sum of these scores was then divided by the maximum possible score (i.e., number of questions multiplied by 2), with 1 indicating perfect performance and 0 indicating participants answered each question incorrectly twice before selecting the correct answer.
3. Catch trials. To verify participants' attention during the task, two catch trials were displayed in which participants were prompted to click on an image of a fish or a can. Participants who failed at least one catch trial were excluded from all analyses. Two children failed one of the catch trials and were thus excluded from the analyses.
4. Expected value (EV) comparison check. Upon completion of the information-seeking task, participants were asked to complete 10 trials which measured the children's ability to compare the expected value of potential outcomes. Participants were presented with ten trials from the information-seeking task and were asked to indicate which pond was better (see Supplementary Methods for instructions). We calculated the proportion of trials each participant answered correctly (that is, selected the pond with the higher EV).
5. Fishing choices check. To further assess participants' understanding of the task, we measured the proportion of correct "fishing" choices—that is, the proportion of times a participant chose to "fish" when the expected value of that pond was positive and not to fish when negative (either option was counted as correct when the expected value was zero).

**Analysis.** For each pond, we calculated (1) the expected value of the pond (EV is equal to the average of the value of items, with big fish = 20 points; small fish = 10 points; large can = −20 points; small can = −10 points, seaweed = 0 points), (2) the number of seaweed (varied from 0 to 6) and (3) the likelihood that the participant will be given the possibility to decide whether to fish ("agency" could be equal to 0%, 50% or 100%). Then for each trial, we calculated the difference between the two ponds on each of the three measures above (right pond − left pond). This gave us (1) ΔEV, (2) Δuncertainty and (3) Δagency, respectively. These Δs were not correlated with each other. In particular, there was no significant correlation between ΔEV and Δagency (Experiment 1: $r(6118) = 0.007$, $p = 0.604$, 95% CI = [−0.03, 0.02]; Experiment 2: $r(5878) = −0.004$, $p = 0.737$, 95% CI = [−0.03, 0.02]), Δuncertainty and Δagency (Experiment 1: $r(6118) = −0.003$, $p = 0.02$, 95% CI = [−0.03, 0.02]; Experiment 2: $r(5878) = −0.007$, $p = 0.604$, 95% CI = [−0.03, 0.02]), ΔEV and Δuncertainty (Experiment 1: $r(6118) = 0.008$, $p = 0.507$, 95% CI = [−0.02, 0.03]; Experiment 2: $r(5878) = 0.01$, $p = 0.316$, 95% CI = [−0.01, 0.04]) across trials. Δs fluctuations across trials for an example participant are shown in Supplementary Fig. 1.

To assess the relative impact of EV, agency, and uncertainty on participants' information-seeking choices, we ran a logistic mixed-effects regression predicting information-seeking choices on each trial (coded as 1 if the participant selected right pond and 0 if they selected left. The following factors were included as fixed effects and random slopes for each participant: ΔEV (right-left), Δuncertainty (right-left), Δagency (right-left). We also included fixed effects of instructions comprehension score, EV comparisons task score, and correct fishing choices as covariates. To determine the developmental trajectory of each motive, the model also included age in years and its interaction with each of the three motives. Results remain consistent if age is log-transformed to account for potential non-linear patterns and/or measured in months rather than years. Three additional models were run. One model was similar to the one described above but with the inclusion of a factor representing information-seeking choices made on the previous trial (0 for left, 1 for right). A second model was run on the same data as the first one but included triple interactions between age and each pair of information-seeking motives (ΔEV, Δuncertainty, and Δagency). A third model, similar to the main one described above, was run on trials in which Δagency was equal to 0, i.e., trials in which the agency probability was identical for the two sides of the screen. The intention here was to test whether the effects of ΔEV and Δuncertainty would remain significant in the absence of agency. Therefore, this model did not include Δagency or its interaction with age as predictors.

As complementary analyses, we fit the children's data at the individual level by estimating their β coefficients for ΔEV, Δagency, and Δuncertainty from information-seeking choices on a trial-by-trial basis.

Regularized ridge regression was conducted with the glmnet R package[57] (version 4.1–4) to avoid overestimating the coefficients. Individual participants' coefficients were then correlated (via Pearson correlation) with participants' age to further assess the relationship between the effects of each information-seeking motive and developmental stage. The BayesFactor R package[58] (version 0.9.12-4.3) was used to calculate Bayes factors. All default options were used, including non-informative Jeffreys's priors[59].

To ease interpretability and comparison among predictors[60], all numeric variables were rescaled using the R scale function[61], which divides a variable by its standard deviation. For ΔEV, Δuncertainty, and Δagency the center argument was set to false to preserve the original sign, in accordance with the fact that values of 0 were meaningful for these variables[62]. For all other numeric variables, the center argument was set to true so that predictors would have a mean of 0, essentially z-scoring those variables. None of our analyses were affected by rescaling. For illustrative purposes, age was converted back to raw values by inverting the z-score transformation (Figs. 2 and 3). Models were implemented in R with the lme4 package[63] (version 1.1–29), which was used to obtain the statistical estimates for each effect. Additionally, we tested the significance of each effect of interest by performing likelihood ratio tests using the full model and a nested model missing the target factor. For these tests, the R (version 4.2.0) anova function was used[61].

For illustrative purposes, we plotted the estimated probability of seeking information on the right side as a function of each of the three key variables while keeping the other variables constant for each age (Fig. 2) using the interactions R package[64] (version 1.2.0). The Johnson-Neyman technique[40] was used to estimate the age at which the effects of each information-seeking motive (ΔEV, Δuncertainty, and Δagency) became significant ($p < 0.05$). These values were estimated from the same model as described above using the modelbased R package[65] (version 0.8.5). To ease interpretability, we report the age in years instead of z-scored age in years, at which effects became significant.

The data met the assumptions of the statistical tests used. Where applicable, normality and equal variances were formally tested, and when assumptions of normality were violated, non-parametric tests were conducted.

Custom data analysis codes are provided at https://github.com/affective-brain-lab/information-seeking-children/.

## Experiment 3
**Participants.** Twenty-nine children completed the experiment online. Two participants failed one of the attention checks. Thus, data from 27 children were analyzed (age: $M = 6.93$ years ± 2.6 years, age range = 4–12 years, 59.3% female, 40.7% male). Consent, assent, and ethical approval were obtained as in the main experiment. $N$ was based on the main experiment.

**Procedure.** The task was similar to the main task. However, only ten trials were displayed, and each contained the same combinations of items across participants. Trials were constructed such that in one pond, the likelihood of participants' agency was always 100% and in the other pond always 50%. The two ponds displayed on each trial were otherwise identical (Supplementary Fig. 2). The full instructions and comprehension task are reported in the Supplementary Methods. We did not include any seaweed stimuli in this task.

**Analysis.** A "Δagency" factor was calculated as outlined above for the main experiment. A logistic mixed-effects model was run to estimate the effects of Δagency on children's information-seeking choices. The Δagency factor was also included as a random slope. The model was thus similar to the one run in the main experiment, but because in all the 10 trials of this task, uncertainty levels were zero and EV was held

constant between the two sides (i.e., the Δ between right and left was zero), factors related to uncertainty and EV were not included in the model. The same rescaling and centering procedures were applied as in the main experiment. Significance tests were performed through model selection via chi-square difference tests as detailed for the main experiment.

## Experiment 4
**Participants.** Forty-one children completed the experiment online. One participant failed one of the attention checks. Thus, data from 40 children were analyzed (age: $M = 7.45$ years ± 2.57 years, age range = 4–12 years, 42.5% female, 57.5% male). Consent, assent, and ethical approval were obtained as in the main experiment. $N$ was calculated based on the main experiment.

**Procedure.** The task was similar to the one presented in Experiment 3. However, on each trial, the EV and standard deviation of the visible items were the same across ponds, and on both sides, the fisherman was guaranteed to make the fishing choice, while one pond contained a higher number of "full seaweeds" than the other (Supplementary Fig. 3). The total number of seaweeds (i.e., "full seaweeds" plus "empty seaweeds") was the same on each pond. The full instructions and comprehension task are reported in the Supplementary Methods.

**Analysis.** A "Δuncertainty" factor was calculated as outlined above for the main experiment, without counting "empty seaweeds". A logistic mixed-effects model was run to estimate the effects of Δuncertainty on children's information-seeking choices. The Δuncertainty factor was also included as a random slope. The model was thus similar to the one run in Experiment 3, but because in all the 10 trials of this task EV and agency were held constant between the two sides (i.e., the Δ between right and left was zero), factors related to EV and agency were not included in the model. The same rescaling and centering procedures were applied as in the main experiment. Significance tests were performed through model selection via chi-square difference tests as detailed for the main experiment.

## Experiment 5
**Participants.** Fifty-five children completed the experiment online. One participant failed the instructions comprehension task and was therefore excluded from all further analyses. Thus, data from 54 children were analyzed (age $M = 7.24$ years ± 2.49 years, age range = 4–12 years, 51.85% female, 48.5% male). Consent, assent, and ethical approval were obtained as in the main experiment. Required $N$ for a power of 80% and an alpha of 0.05 was calculated based on the main experiment.

**Procedure.** In each of the ten trials, participants viewed two fishermen sitting on two sides of a river (Supplementary Fig. 4). They were then asked whether they thought each fisherman was going to catch a fish or a can and how sure they were about their response (the order of ponds questioned was randomized across trials). They reported their confidence on the scale shown in Supplementary Fig. 4.

Uncertainty was defined in the main task as the amount of seaweed. We randomized the side on which the pond with higher uncertainty was displayed. Differences in uncertainty between the two sides covered the same range that participants viewed in the main task. Participants completed an instructions comprehension task before proceeding to the main part of the experiment (see below for all questions asked in this experiment).

**Analysis.** For each trial, we calculated the difference in uncertainty (Δuncertainty) between the two sides of the screen (i.e., the number of seaweeds in the right pond minus the number of seaweeds in the left pond). We then calculated the difference in confidence (Δconfidence)

between the two sides (i.e., the self-report for the right side minus the self-report for the left side).

A logistic mixed-effects model was run to predict Δconfidence from Δuncertainty (entered as fixed effect and random slope for each participant), scores on the instructions comprehension task (as fixed effect), and age and its interaction with Δuncertainty (as fixed effect). The same rescaling and centering procedures were applied as in the main experiment. Significance tests were performed through model selection via chi-square difference tests as detailed for the main experiment. The Johnson-Neyman technique[40] was used to estimate the age at which the effects of Δuncertainty on Δconfidence became significant.

Additionally, we tested whether accounting for uncertainty tracking could explain differences in sensitivity to Δuncertainty across ages in the main experiment. To that end, we calculated the correlation between Δconfidence and Δuncertainty across participants separately for each age and included the Pearson R as covariates in the main experiment. This allowed us to control for age-specific uncertainty tracking ability in the main experiment based on measurements from Experiment 5.

**Experiment 6 and Experiment 7**

**Participants.** Thirty adults took part in Experiment 6, and 29 adults took part in Experiment 7. All participated online through Prolific (https://www.prolific.co/) and received a fixed compensation of £7.50. In Experiment 6, the data of two participants were lost due to technical issues with the online host. All other participants passed soundchecks, catch trials and instructions comprehension tests. Thus, data from 28 adults were analyzed in Experiment 6, and data from 29 participants were analyzed in Experiment 7 (Experiment 6: age $M = 28.6 \pm 11.50$ years, age range = 18–66, 32.1% female, 64.3% male, 3.6% other; Experiment 7: age $M = 40.2 \pm 11.70$ years, age range = 22–64 years, 34.5% female, 65.5% male). 42% of participants reported their education level. Of those, 0.04% had less than a high school diploma; 0.04% had no more than a high school degree; 0.16% had at most a 2-year degree; 36% had at most a 4-year degree; 40% had a professional degree (such as a nursing degree, a technician degree, etc.) and/or a graduate degree.

**Procedure.** The procedure was identical to the one described for children, except participants gave consent for themselves. The University College London Research Ethics Committee approved these studies.

**Analysis.** The analysis was the same as described in the main study, except that age was not included as a factor.

**Reporting summary**
Further information on research design is available in the Nature Portfolio Reporting Summary linked to this article.

# Data availability
All data have been made publicly available at a dedicated GitHub repository[66] (https://github.com/affective-brain-lab/information-seeking-children/), which has also been released on Zenodo (https://zenodo.org/record/8151641). Raw data have been anonymized and consolidated into a smaller number of files to increase readability; information in html strings regarding the stimuli presented was transformed into measures of interest (EV, uncertainty, etc.).

# Code availability
All data analysis scripts have been made publicly available at a dedicated GitHub repository[66] (https://github.com/affective-brain-lab/information-seeking-children/), which has also been released on Zenodo (https://zenodo.org/record/8151641).

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

## Acknowledgements

This work was funded by a Wellcome Trust Senior Research Fellowship 214268/Z/18/Z to T.S. We thank Bastien Blain, Christopher Kelly, Laura Globig, Moshe Glickman, Sarah Zheng, and Valentina Vellani for comments on previous versions of the manuscript, and Leo McDermott and Livia McDermott for help with task design.

## Author contributions

T.S. and I.C.D. developed the study concept. T.S., I.C.D., and G.M. contributed to the study design. Data collection was performed by G.M., S.K.B., and C.M.; G.M. performed the data analysis and interpretation under the supervision of I.C.D. and T.S.; G.M., T.S. and I.C.D. drafted the manuscript. All authors approved the final version of the manuscript for submission.

## Competing interests

The authors declare no competing interests.
