## [Peer Review File · Nature Communications]

Reviewers' Comments:

Reviewer #1:

Remarks to the Author:

This is an interesting paper that tackles information search across childhood, and compares this with a small group of adults. The task is innovative but also complex. The results suggest that the tendency to seek good news is stable throughout development, while the motivation to reduce uncertainty or increase utility remains increasing across childhood. What is also very impressive is that the authors replicated their results in an independent sample. However, from a developmental perspective there are a few issues with the current task and dataset that limit the interpretation and scope of these results.

The task is highly complex, even for the reader it was at this point, even with the figures, hard to follow what the participant needs to keep track of. Seeing the task I think it is truly amazing that the authors made kids 4-6 able to even score above chance level. It is also visible that in the comprehension checks that young kids are worse but able to replicate most of the instructions. However, that does not mean that during the task they are able to manage all the moving parts. As we know from the developmental literature children and also older people with less mental capacities available will often use heuristics to solve complex tasks, heuristics that fit their mental abilities. And it is extremely likely that they apply those also in this complex task. For instance they would just go for the good news heuristic. Or sometimes just explore things. This is actually something different from trading of different motivations for information search, and this would also suggest that what you will find in the data is not so much linear increases in the weight that is put on the different motives but rather differences in frequency use of different heuristics.

The complexity of the task might use very well for adults and the narrative here is suitable for kids but it is not clear if the task is just too complicated and the difference is that kids perform at the edge (or beyond) of their ability and adults don't, resulting in different strategies. Not sure if this caveat can really be fully addressed but, on more additional analyses of interest would be including correct fishing choices as an additional variable to your model, because it another good indicator of how difficult the task really is.

It would also be interesting to see how the developmental differences in strategies result in actually differences in rewards. In these types of stochastic tasks we have previously seen, like in the two step task, that the lower cognitive effort strategy of model-free learning yielded similar rewards as the model based one. In that sense the "simpler" behavior of kids may be most adaptive.

Related to that I was somehow expecting computational modeling, given that it was suggested that a framework was developed for this task. That could also help with running simulations and relate strategies with rewards and see how that changes across development. Adding such a framework would enrich the paper in many ways and would make it probably also interesting to a wider audience. (The other thing is trying to make the discussion a bit broader in terms of what this may use for instance for education practices, and linking it even a bit more back to all the old developmental work on search on information boards, there is a quite a bit).

Next big issue are the way the data is analysed. First of all it is not clear why the data are not analysed as a continuous variable which it clearly is. The current binning and subsequent analyses that include adults as just one age bin rely on the rather strange assumption that distance between each bin is equal, or looking at it from another way that they may all be very different.

From the start it is not clear why the adults are actually needed in the comparison, but if so it should be done appropriately. Also in the graphs are currently misleading in suggestion that 18 is very close to 12 while it is not.

It would be good to start analysing the data of the children separately, and also use age as a continuous variable (then one can still look for non-linear trends in many different ways).

If you see the developmental trends during childhood it does not seem to make a lot of sense to directly compare all children with all adults.

Finally, for a developmental paper it would be valuable to discuss what the underlying developmental mechanism is or could be. This ideally should have driven the design of the task and the hypotheses. Because it is somewhat trivial that we will see increases in performance on complex tasks, and that strategies will also become more complex and will integrate more features of the task. But in this case, what are the mechanisms? Is it related to working memory? Is it related to switching ability etc. The advantage of starting at this point is that you could have included such measures.

Related, it is quite standard in developmental psychology to get some measures of intellectual ability, like IQ, in order to make sure that cognitive ability is evenly distributed across your samples. It is very well possible that your adult population are high IQ students and the kids are all average IQ kids. This is a potentially huge confound.

Reviewer #2:

Remarks to the Author:

In this paper, Molinaro et al. investigate how different aspects of information influence information-seeking behavior from age 4 to age 12, and in adults. Using a child-friendly information-seeking task, they find that the influence of agency and uncertainty increases across development, while the influence of expected value remains consistent across age. I believe the researchers set out to answer a very interesting question with an impressively large sample size. I also appreciate their inclusion of a replication sample. I did, however, have a number of concerns about their interpretation of their analyses, which limit my enthusiasm for the manuscript in its present form. I think some of these concerns can be addressed in a revision.

My main concern is in how the three variables of interest (delta EV, delta agency, and delta uncertainty) were treated in the analyses of information-seeking decisions and interpreted. Interactions between the three effects were not modeled, but the resolution of uncertainty and the desire to reveal positive information seem like they may interact with 'agency' in important ways. Thus, I think it is important to understand the age x agency x uncertainty and the age x agency x EV interaction effects. For example, resolving uncertainty has instrumental utility on agency trials. Thus, to understand developmental change in uncertainty-seeking behavior, it seems important to differentiate the motivation to resolve uncertainty for the sake of resolving uncertainty (e.g., the effect of delta uncertainty on trials in which delta agency was 0 or, especially on trials where the fisherman made the choice for both ponds) and the motivation to resolve uncertainty because it will be helpful in improving one's choices (e.g., the effect of delta uncertainty on trials in which delta agency was > 0.) Along similar lines, the effect of delta EV was interpreted as a motivation to reveal positive information, but it seems important to understand age-related change in revealing positive information when you can then decide whether to choose it vs. positive information about the fisherman's choice. The 'pure' effect of wanting to receive good news across age groups is potentially interesting but not currently reported anywhere (i.e., not just when delta agency is 0, but when both ponds are controlled by the fishermen only).

Did the extent to which participants' considered the instrumental value of information actually improve their subsequent fishing choices? This would also be a helpful manipulation check to demonstrate that this information did indeed have instrumental value.

The authors had nice control analyses to convincingly demonstrate that younger children understood the expected value manipulation. But did they understand the uncertainty manipulation? Previous research has found a stronger influence of uncertainty at younger ages, even when multiple motives (e.g., information vs. reward) compete (e.g., Blanco & Sloutsky, 2021). Thus, these results are somewhat in conflict with the literature, and may suggest that they did not compute or track uncertainty in this task.

Finally, it is somewhat concerning (though perhaps not surprising) that the influence of all

variables either remains consistent, or strengthens with age. This seems to suggest that younger participants were making more random choices. It would be helpful to know whether participant reaction times increase or decrease with age. Increases with age would be more concerning as they may just suggest that young children are not really considering much when making their info-seeking decisions.

More minor concerns:

I found the description of the task somewhat difficult to follow. It would be helpful to state participants' goal up front: to make the best 'fishing' choice for 0, 1, or 2 ponds on each trial, which they could do by revealing information about both ponds. Then it would be helpful to state that although participants' goal was to make the best fishing choice, the analyses of interest focused on the information choice at the start of the trial. Also, were participants incentivized to perform well?

I also found the giant lists of statistics for many of the analyses very difficult to parse. The tables included in the supplement are much easier to process, and I think most of the results should be reported in these compact tables rather than long lists with each age group and each experiment.

It is not clear why gender was included in the model. Including control variables without sufficient justification can lead to biased estimates (see Wysocki et al., 2022).

The methods and results are also reported incorrectly in several places. For both studies, the age range is reported as 10 - 12 and 10-11, and not 4 - 12. In addition, the description of the models state that delta EV, agency, and uncertainty are included in the model as both fixed and random intercepts. This does not really make any sense, but in looking at the code (thank you for providing that on github!) it seems as though the authors actually included delta EV, agency, and uncertainty as fixed effects and random slopes for each participant, which makes a lot more sense.

Reviewer #3:

Remarks to the Author:

In this paper, the authors test children and adults on an information-seeking task in which three key aspects of information (affective, cognitive and instrumental) are experimentally manipulated. In this way, authors can tear apart the specific aspects children focus on when seeking information, and when they start to do so across development. They find that preschoolers are already sensitive to the affective and instrumental components of information, while they become sensitive to the cognitive component (i.e., the level of uncertainty) only from 6 years of age onwards. The second experiment is a direct replication of the first one, and it finds very comparable results, which shows replicability of the findings.

This paper is quite interesting, it focuses on a trending topic tackling it with a novel paradigm, and produces important findings that will help the field move forward. The methodology is very sound, and the additional experiments and explanations about how the variables were manipulated make it even more robust, although it could make some key improvements on the statistical analysis part.

Having said that, I think the paper in its current form makes a minority of claims that are not directly following from the evidence it provides, and this prevents the paper from being accepted in its current form. I address this below. The comments are not in order of importance, but as a heads-up, the two key points concern statistical analyses (the evolution of the effects across time can be investigated in a more convincing manner) and the interpretation (which partly follows from the limits of the current statistical analyses).

Title and line 24: "How children decide", "how children integrate". I think this study is more about the "what" than the "how". It gives us very crucial insights on what children focus on, but it is not mechanistic: it doesn't get to the cognitive or brain processes that support children's decisions. For this reason, I am not sure that it directly answers a "how" question.

55: To my knowledge, there is no evidence of active information seeking in newborns (for information seeking abilities in infants, see Hunnius, 2022). The evidence cited concerns the second year of life; Please revisit the statement accordingly.

97: "computationally". This word is used here and a couple of other times across the paper (e.g., line 356). It is not immediately clear to me how the study is computational – it is disentangling different motives experimentally, but I'm not sure it does so computationally. I would either justify the claim or modify it.

98: The authors wrote: "Rather, motives have either been studied in isolation or in a situation where they are confounded". Please provide references.

129: Sentence: "Vice versa if they selected the left pond". Please rephrase.

136, 138, and 139: "varied by varying". Please rephrase.

156: "the left pond of the pond": please clarify.

164: Children were split in age groups, and groups were treated as a factor, but I think there would be an advantage in keeping age as a continuous variable. I would suggest a slightly different analysis because the current one also creates problems in the interpretation of the results (for example, it is not possible to conclude that any group has higher or lower beta scores without conducting posthoc tests, see comments below for more information).

Thank to the authors I was able to access the data and the analysis scripts. I introduce the steps I took below, followed by an explanation. I post this script here simply because it is the most efficient way to make my point clear to the authors – this analysis can also be done in other ways, but I do think it comes with many advantages.

- I imported the data with:

```
dat_compt <- read.csv("data_main_experiment_children_1.csv")
```

- Run their script analysis_main_experiment.R

- Run these additional lines at the end of the script (explanation follows):

```
#####
```

```
library(interactions)
```

```
library(modelbased)
```

```
dat_compt2 = dat_compt[c("info_choice", "delta_EV", "delta_uncertainty_level", "delta_agency",  
"age_in_months", "percent_comprehension", "wob", "gender_coded", "subject_ID")]
```

```
dat_compt2 = na.omit(dat_compt2)
```

```
# run the model
```

```
compt_mod <- glmer(info_choice ~ (delta_EV + delta_uncertainty_level +  
delta_agency)*log(age_in_months) +
```

```
percent_comprehension + wob + gender_coded +  
(delta_EV + delta_uncertainty_level + delta_agency  
| subject_ID),
```

```
data = dat_compt2, family = binomial, control = glmerControl(optimizer = "bobyqa",  
optCtrl=list(maxfun=2e5)), nAGQ = 0)
```

```
summary(compt_mod)
```

```
# plot interaction
```

```
interact_plot(compt_mod, pred = delta_agency, modx = age_in_months, interval = TRUE,  
plot.points = FALSE, int.width = 0.8, line.thickness = 1, modx.values = c(54, 78, 102, 126, 150 ))
```

```
# estimate predictive effects
```

```
slopes <- estimate_slopes(compt_mod, trend = "delta_agency", at = "age_in_months", length =  
54)
```

```
# plot significance of the effect across time (Johnson-Neyman plot)
```

```
plot(slopes)
```

```
# show stats
```

slopes

#####

In the model, time is interacting with the main variables of interest (the deltas). I added a logarithmic function because it seems a sensible idea that these abilities settle at some point across development (and this was also visible from the figures in the paper). When plotting the effects, you can see how they evolve across development. For this reason, I think this kind of analysis is better suited to studying developmental change. When plotting the slopes with a Johnson-Neyman plot, you can also identify the precise moment across development when the effect becomes significant.

170: Comprehension scores have been clustered all together, but they concern very different aspects of the task (understanding of the instructions, theory of mind...). Is there any specific aspect that is affecting the results (vs overall understanding)? I'd like to see the descriptive stats of each question separately – in the response, not necessarily in the paper.

210: The title does not reflect the content of the chapter.

226: What the standard deviation of the items is could be made a bit clearer here, even if it is addressed in methods.

249: To my knowledge, logistic regression works with the likelihood ratio chi-square test. Is this the same as the chi-square difference test?

271: $p = 0.015$ instead of 0.15.

274 "Together, these results suggest that information-seeking choices that help regulate emotion remain stable throughout development."

This conclusion of stability is inferred from the lack of an effect. Moreover, if the ability is gradually increasing, testing binned age as independent groups will not pick it up. The analysis I suggest above would also help with this: If the beta coefficients for interaction between time and main independent variables are significantly different from zero, there is developmental change. If they are not different from zero, it is still unwarranted to make conclusions about stability, unless Bayesian stats and Bayes factors for the null hypotheses are reported (or at least effect sizes for frequentist tests).

280: "which becomes even greater". How has this been tested? Similar to the previous point for line 274.

291 and 307: You may need posthoc tests or analysis as described in my comment for line 164 if you want to better characterize this increase.

498: the correlations are reported and I was convinced that the independent variables are not correlated with each other – which is a really good aspect of the design. As a suggestion – It would be more convincing to have a figure with an example sequence of trials, with trial number on the x-axis and values of the three main dependent variables on the y-axis.

Final comment:

The theoretical framework is robust, supported by evidence, and explains the data well. However, in the current formulation of the paper, it might come across that the framework is also complete, in the sense that all kinds of information seeking can be explained through it. However, some drives for information are partly unrelated, especially coming from the developmental literature, where infants' information seeking has traditionally been related to novelty/complexity (see Kidd & Hayden, 2015 for a review) and more recently to learning progress (see Poli, Serino, Mars & Hunnius, 2020). Learning progress has also been related to adults' exploratory behaviour in very recent studies (Poli, Meyer, Mars, & Hunnius, 2022; Ten, Kaushik, Oudeyer, & Gottlieb, 2021). These drives cannot be addressed within the current study; For example, there is no opportunity to learn or to get better at the task, which makes it difficult to study learning progress; the elements never change, which makes it difficult to study novelty. I would either explain how that

literature relates to the current theory (are all of these different aspects of the cognitive part of this theory, maybe?), or at least acknowledge that more drives than the ones addressed here exist.

Related to this point, the current theory succeeds in capturing the multifaceted nature of information seeking. The breadth of its aim is excellent, but future research will need to move towards greater depth as well. For example, once we've acknowledged that information is driven by cognitive aspects, we need to determine how exactly this happens, and what are the mechanisms that underlie this information seeking: for example, an attunement to learning progress might be the policy that allows agents to maximize information (see Oudeyer's research). This is not a limitation of the current study – The study is robust and it delivers what it promised (which is a lot, in my opinion). Rather, this is a point for future research that I believe it's worth mentioning.

References

Hunnius, S. (2022). Early cognitive development: five lessons from infant learning. In Oxford Research Encyclopedia of Psychology.

Kidd, C., & Hayden, B. Y. (2015). The psychology and neuroscience of curiosity. *Neuron*, 88(3), 449-460.

Poli, F., Serino, G., Mars, R. B., & Hunnius, S. (2020). Infants tailor their attention to maximize learning. *Science advances*, 6(39), eabb5053.

Poli, F., Meyer, M., Mars, R. B., & Hunnius, S. (2022). Contributions of expected learning progress and perceptual novelty to curiosity-driven exploration. *Cognition*, 225, 105119.

Ten, A., Kaushik, P., Oudeyer, P. Y., & Gottlieb, J. (2021). Humans monitor learning progress in curiosity-driven exploration. *Nature communications*, 12(1), 1-10.

We were encouraged by the assessment of the reviewers that the paper “produces important findings that will help the field move forward. The methodology is very sound, and the additional experiments and explanations about how the variables were manipulated make it even more robust.” (Reviewer 3); that “the researchers set out to answer a very interesting question with an impressively large sample size” (Reviewer 2); and that “what is also very impressive is that the authors replicated their results in an independent sample.” (Reviewer 1).

The Reviewers also provided helpful suggestion and comments, all of which we were able to incorporate and address in the revised manuscript. In particular, as suggested by both Reviewer 1 and 3 we reanalyzed the data using age as a continuous variable. The new analyses confirm and strengthen our previous results. The results also hold after including the various covariates suggested by Reviewer 1 and 2. Importantly, we conducted an additional control experiment (N = 54) suggested by Reviewer 2, which shows that kids as young as 4 track uncertainty in our task.

We would like to thank the reviewers for a thorough and thoughtful commentary on our work. Incorporating their suggestions has strengthened the manuscript which we hope is now suitable for publication in *Nature Communications*. To ensure we address all the Reviewers’ comments, and for ease of reference, we include the reviews below (in bold) followed by our response to each concern.

Reviewer #1

This is an interesting paper that tackles information search across childhood, and compares this with a small group of adults. The task is innovative but also complex. The results suggest that the tendency to seek good news is stable throughout development, while the motivation to reduce uncertainty or increase utility remains increasing across childhood. What is also very impressive is that the authors replicated their results in an independent sample. However, from a developmental perspective there are a few issues with the current task and dataset that limit the interpretation and scope of these results.

- We thank the reviewer for this evaluation. We now address all the reviewer’s concerns and follow the reviewer’s helpful suggestions.

The task is highly complex, even for the reader it was at this point, even with the figures, hard to follow what the participant needs to keep track of. Seeing the task I think it is truly amazing that the authors made kids 4-6 able to even score above chance level. It is also visible that in the comprehension checks that young kids are worse but able to replicate most of the instructions. However, that does not mean that during the task they are able to manage all the moving parts. As we know from the developmental literature children and also older

people with less mental capacities available will often use heuristics to solve complex tasks, heuristics that fit their mental abilities. And it is extremely likely that they apply those also in this complex task. For instance they would just go for the good news heuristic. Or sometimes just explore things. This is actually something different from trading of different motivations for information search, and this would also suggest that what you will find in the data is not so much linear increases in the weight that is put on the different motives but rather differences in frequency use of different heuristics.

- The reviewer raises an interesting possibility – that younger children use heuristics to solve the problem. To examine this, we run two new analyses. First, we directly test the reviewer’s prediction that if this is indeed the case, we will *not* find a linear increase in the weight that is put on the different motives with age. To test this, we fit regularized logistic regression models separately for each child, using information-seeking choices as the dependent variable and ΔEV , Δ uncertainty, and Δ agency as the predictors of interest. This provided us with betas for each participant. We then correlated individual participants’ beta estimates for the three predictors of interest with age across all children. The results are consistent with our original finding, the effects of Δ uncertainty (Experiment 1: $r(202) = 0.28$, $p < 0.001$; Experiment 2: $r(193) = 0.19$, $p = 0.007$) and Δ agency (Experiment 1: $r(202) = 0.32$, $p < 0.001$; Experiment 2: $r(193) = 0.21$, $p = 0.003$) both significantly increase with age across children, while the effects of ΔEV does not (Experiment 1: $r(202) = 0.09$, $p = 0.200$; Experiment 2: $r(193) = 0.01$, $p = 0.856$). These results suggest that there is indeed a linear increase in the weight that individuals put on the different motives across development. We now include this analysis in the revised manuscript (pg. 10).
- We were, however, inspired by the reviewer’s insight and decided to test an even simpler heuristic. Namely, we tested whether children simply repeat the information-seeking choice made on the previous trial. To that end, we ran a mixed effects logistic model predicting information-seeking choice from age, Δ uncertainty, ΔEV , Δ agency, and a heuristic – namely, choice on last trial – as well as the interaction of each of these factors with age. In Experiment 1, there was a significant interaction between age and ‘previous choice heuristic’ ($\chi^2(1) = 9.51$, $p = 0.002$), suggesting that younger children used this heuristic more than older ones. However, this interaction was not replicated in Experiment 2 ($\chi^2(1) = 2.06$, $p = 0.151$). Importantly, even after accounting for this heuristic we still observed, in both experiments, an interaction between age and Δ uncertainty (Experiment 1: $\chi^2(1) = 23.96$, $p < 0.001$; Experiment 2: $\chi^2(1) = 9.27$, $p = 0.002$) and between age and Δ agency (Experiment 1: $\chi^2(1) = 23.92$, $p < 0.001$; Experiment 2: $\chi^2(1) = 9.03$, $p = 0.003$), and no interaction between age and ΔEV (Experiment 1: $\chi^2(1) = 3.01$, $p = 0.083$, Experiment 2: $\chi^2(1) = 0.16$, $p = 0.686$). Thus, while in Experiment 1 younger children used a simple heuristic to solve the problem more than older children, in both experiments accounting for this heuristic did not alter the main results. These new analyses are now reported in the revised manuscript (pg. 13).

- Regarding complexity – we appreciate that the task seems especially complex on paper, but in fact when you engage with it in real-time with the narrator and examples it is not as complex as it seems on the page. This is partially because we provide a lot of information in the manuscript for the benefit of other scientists, which the subjects themselves are unaware of. When developing the task, we piloted it on 4- and 6-year-olds. They followed the game easily and enjoyed playing. We believe it is as easy as the average digital game children play online. We now clarify this in the manuscript (pg. 17).

The complexity of the task might use very well for adults and the narrative here is suitable for kids but it is not clear if the task is just too complicated and the difference is that kids perform at the edge (or beyond) of their ability and adults don't, resulting in different strategies. Not sure if this caveat can really be fully addressed but, one more additional analyses of interest would be including correct fishing choices as an additional variable to your model, because it another good indicator of how difficult the task really is.

- This is an excellent suggestion. We now include correct fishing choices as a covariate in all our analyses. Even accounting for this factor, we find that all our results remain. For example, we run a mixed linear model (with fixed and random effects) predicting information-seeking choice from age, Δ uncertainty, Δ EV, Δ agency, correct fishing choices, comprehension scores and the interaction of each of these factors with age. Even after accounting for correct fishing choices we still observed an interaction between age and Δ uncertainty (Experiment 1: $\chi^2(1) = 23.85$, $p < 0.001$; Experiment 2: $\chi^2(1) = 10.33$, $p = 0.001$) and between age and Δ agency (Experiment 1: $\chi^2(1) = 24.19$, $p < 0.001$; Experiment 2: $\chi^2(1) = 10.12$, $p = 0.001$), and no interaction between age and Δ EV (Experiment 1: $\chi^2(1) = 2.86$, $p = 0.091$; Experiment 2: $\chi^2(1) = 0.14$, $p = 0.710$). We now include this control analysis in the revised manuscript (pg. 9-13).

It would also be interesting to see how the developmental differences in strategies result in actually differences in rewards. In these types of stochastic tasks we have previously seen, like in the two step task, that the lower cognitive effort strategy of model-free learning yielded similar rewards as the model based one. In that sense the “simpler” behavior of kids may be most adaptive.

- To examine the relationship between rewards and age we calculated the reward on each trial and then averaged the rewards for each participant. Older age was associated with greater rewards (Experiment 1: $r(202) = 0.20$, $p = 0.005$; Experiment 2: $r(194) = 0.27$, $p < 0.001$).
- Note, that following previously published similar tasks in adults (Cogliati Dezza et al., 2022; Kelly & Sharot, 2021), the outcome (fish/can) on each trial are never revealed to the subject. This is crucial, because these tasks are meant to model real world situations where people seek information that provide clues about future outcomes (e.g., what gift am I likely to get for my birthday? how likely am I to get cancer?), but where those outcomes will only be known in the far future (and sometimes never). The reason this is done is that

if the outcomes were revealed immediately there would be very little non-instrumental utility for advanced information in the form of reduced uncertainty and positive affect (for detailed explanations see for example Cogliati Dezza et al., 2022). We now clarify this in the revised manuscript (pg. 5).

Related to that I was somehow expecting computational modeling, given that it was suggested that a framework was developed for this task. That could also help with running simulations and relate strategies with rewards and see how that changes across development. Adding such a framework would enrich the paper in many ways and would make it probably also interesting to a wider audience.

- Following the reviewer's comment it became apparent to us that we did not clarify the concept of "rewards" in this task and how they relate to information-seeking strategies. Our theoretical (Sharot & Sunstein, 2020) and empirical (Cogliati Dezza et al., 2022; Charpentier et al., 2018; Kelly & Sharot 2021) approach to information-seeking is different from the traditional approach, in that we suggest humans (and non-humans) are not simply trying to maximize 'external rewards' when seeking information. Rather they are also trying to maximize 'internal rewards' in the form of positive feelings and reduction of uncertainty, even when information is non-instrumental. Thus, we compute not only external rewards and punishment (fish and cans) but also internal rewards. We have recently published a paper in which we do exactly that in adults (Cogliati Dezza et al., 2022) and we now use the same approach here. We calculated the three reward types (fish/cans obtained after fishing; uncertainty reduction; increase in positive information) and normalize them within reward type to put them on the same scale (between 0 and 1). We find that, on average, young children obtained less rewards than older children (Experiment 1: $r = 0.14$, $p = 0.005$; Experiment 2: $r = 0.23$, $p = 0.001$).
- Our model is a simple linear regression that allows to compute weights assigned to each factor. The weights represent the importance assigned to the three motives. While simple, it is the best representation of our theory, and we explicitly describe it as such here and elsewhere (Sharot & Sunstein, 2020; Kelly & Sharot, 2021; Cogliati-Dezza et al., 2022; Charpentier et al., 2018; Vellani et al., 2020; Vellani et al., 2022). It is possible that the reviewer has a computational model of learning in mind (such as reinforcement learning models), but because outcomes are not revealed in these types of tasks (that is a subject never knows if they fished a can or fish) these models are not used.

(The other thing is trying to make the discussion a bit broader in terms of what this may use for instance for education practices, and linking it even a bit more back to all the old developmental work on search on information boards, there is a quite a bit).

- Thank you for suggesting these interesting points for discussion, which we have now included in our manuscript (pg. 14-15).

Next big issue are the way the data is analysed. First of all it is not clear why the data are not analysed as a continuous variable which it clearly is.

- Following the reviewer's suggestion, we now treat age as a continuous variable. The results remain the same. In particular, we ran a mixed linear model (with fixed and random effects) predicting information-seeking choice of all children from age, ΔEV , $\Delta uncertainty$, $\Delta Agency$, correct fishing choices, comprehension scores and the interaction of each of these factors with age (pg. 9-10). We found an interaction between age and $\Delta uncertainty$ (Experiment 1: $\chi^2(1) = 23.85$, $p < 0.001$; Experiment 2: $\chi^2(1) = 10.33$, $p = 0.001$) and between age and $\Delta Agency$ (Experiment 1: $\chi^2(1) = 24.19$, $p < 0.001$; Experiment 2: $\chi^2(1) = 10.12$, $p = 0.001$), and no interaction between age and ΔEV (Experiment 1: $\chi^2(1) = 2.86$, $p = 0.091$; Experiment 2: $\chi^2(1) = 0.14$, $p = 0.710$). In addition, there were main effects of ΔEV (Experiment 1: $\chi^2(1) = 70.59$, $p < 0.001$; Experiment 2: $\chi^2(1) = 82.89$, $p < 0.001$), $\Delta uncertainty$ (Experiment 1: $\chi^2(1) = 41.59$, $p < 0.001$; Experiment 2: $\chi^2(1) = 46.44$, $p < 0.001$), and $\Delta Agency$ (Experiment 1: $\chi^2(1) = 128.71$, $p < 0.001$; Experiment 2: $\chi^2(1) = 122.23$, $p < 0.001$; pg. 9).
- Note, that for the purpose of verifying instruction comprehension only we did keep age groups. This is simply to make sure each age group comprehend all aspects of the task (pg. 8).

The current binning and subsequent analyses that include adults as just one age bin rely on the rather strange assumption that distance between each bin is equal, or looking at it from another way that they may all be very different. From the start it is not clear why the adults are actually needed in the comparison, but if so it should be done appropriately.

- We take the reviewer's point and now include only children in all our main analysis. We analyze the adults' data separately only to show that their information-seeking decisions are indeed also driven by ΔEV , $\Delta uncertainty$ and $\Delta Agency$. Adults' data are now in the Supporting Information.

Also in the graphs are currently misleading in suggestion that 18 is very close to 12 while it is not.

- Thank you for pointing this out. Given that we no longer include comparisons between children and adults this is no longer the case.

It would be good to start analysing the data of the children separately, and also use age as a continuous variable (then one can still look for non-linear trends in many different ways).

- We agree. As detailed above, in the new manuscript we now focus our main analyses on the children's data only and use age as a continuous variable in our models.

If you see the developmental trends during childhood it does not seem to make a lot of sense to directly compare all children with all adults.

- We agree and we no longer do this.

Finally, for a developmental paper it would be valuable to discuss what the underlying developmental mechanism is or could be. This ideally should have driven the design of the task and the hypotheses. Because it is somewhat trivial that we will see increases in performance on complex tasks, and that strategies will also become more complex and will integrate more features of the task. But in this case, what is the mechanisms? Is it related to working memory? Is it related to switching ability etc. The advantage of starting at this point is that you could have included such measures.

- After reading the reviewer's comment it became apparent to us that we did not make our development questions, predictions and their rationale, clear. We have previously presented a theory suggesting that three motives drive information-seeking: instrumental utility (that is seeking information to improve external outcomes); uncertainty reduction and hedonic utility (that is receiving good news). We have since provided empirical evidence for this theory in adults (Kelly & Sharot, 2021; Cogliati Dezza et al., 2022; Vellani et al., 2022). We predicted that the impact of these motives on information-seeking will follow different developmental trajectories. Our specific predictions were informed by our brain imaging study in which we found that the weight on the hedonic utility motive is associated with representation of information value in the midbrain (VTA and SN) while the weight put on uncertainty reduction is associated with representation of value in the frontal cortex (Charpentier, Bromberg-Martin and Sharot, 2018). As the subcortical regions fully develop earlier than frontal regions (e.g., Johnson, 2001) we hypothesized that the EV motive will appear earlier in development, while the other motives will appear at a later stage. Complexity alone is unlikely to explain the results, because if children were simply incorporating less features than different children would be incorporating different features and development of different motives would not appear to have different trajectories. Moreover, the motive which seems not to alter across age – EV – is, if anything, the most complex of them all. We now clarify this in the revised manuscript (pg. 4-5).

Related, it is quite standard in developmental psychology to get some measures of intellectual ability, like IQ, in order to make sure that cognitive ability is evenly distributed across your samples. It is very well possible that your adult population are high IQ students and the kids are all average IQ kids. This is a potentially huge confound.

- We follow an earlier suggestion by the reviewer and no longer compare adults to children in this study. Thus, IQ differences between children and adults could no longer be a confound. Nevertheless, we went back to our subjects to get a proxy of intellectual ability. In particular, we asked adults to report their level of education, given that education has been shown to correlate strongly with intellectual ability (Colom et al., 2002). We also asked parents/guardians to report their own education level and that of the other parent/guardian if applicable, as a child's intellectual abilities has been shown to correlate

strongly with the education level of their parents (Jackson et al., 2017). We now report these numbers (pg. 16).

- Note that our adult sample was not composed of students, but rather of Prolific users (Experiment 1 mean age = 28.6; Experiment 2: mean age = 40.2). The average distribution of education levels among Prolific users has been reported to be similar to that of the general population (Johnson et al., 2021; Peer et al., 2017; Uittenhove et al., 2022), as is education level of mothers of Lookit participants (Scott & Schulz, 2017).

Reviewer #2

In this paper, Molinaro et al. investigate how different aspects of information influence information-seeking behavior from age 4 to age 12, and in adults. Using a child-friendly information-seeking task, they find that the influence of agency and uncertainty increases across development, while the influence of expected value remains consistent across age. I believe the researchers set out to answer a very interesting question with an impressively large sample size. I also appreciate their inclusion of a replication sample. I did, however, have a number of concerns about their interpretation of their analyses, which limit my enthusiasm for the manuscript in its present form. I think some of these concerns can be addressed in a revision.

- We thank the reviewer for this positive evaluation. We now address all the reviewer's comments in the revision (see details below).

My main concern is in how the three variables of interest (delta EV, delta agency, and delta uncertainty) were treated in the analyses of information-seeking decisions and interpreted. Interactions between the three effects were not modeled, but the resolution of uncertainty and the desire to reveal positive information seem like they may interact with 'agency' in important ways. Thus, I think it is important to understand the age x agency x uncertainty and the age x agency x EV interaction effects.

- This is a very helpful suggestion. We ran a model with additional interactions suggested by the reviewer (age x agency x uncertainty; age x agency x EV; as well as age x EV x uncertainty) – none were found to be significant (all p values > 0.114). All our other effects remain the same: we found significant effects of ΔEV (Experiment 1: $\chi^2(1) = 70.56$, $p < 0.001$; Experiment 2: $\chi^2(1) = 82.67$, $p < 0.001$), $\Delta Agency$ (Experiment 1: $\chi^2(1) = 127.93$, $p < 0.001$; Experiment 2: $\chi^2(1) = 122.03$, $p < 0.001$), and $\Delta uncertainty$ (Experiment 1: $\chi^2(1) = 41.62$, $p < 0.001$; Experiment 2: $\chi^2(1) = 46.76$, $p < 0.001$). As in the main analyses, both $\Delta Agency$ (Experiment 1: $\chi^2(1) = 23.89$, $p < 0.001$; Experiment 2: $\chi^2(1) = 10.12$, $p = 0.001$) and $\Delta uncertainty$ positively interacted with age (Experiment 1: $\chi^2(1) = 24.55$, $p < 0.001$; Experiment 2: $\chi^2(1) = 10.37$, $p = 0.001$), while ΔEV did not (Experiment 1: $\chi^2(1) = 2.82$, $p = 0.093$; Experiment 2: $\chi^2(1) = 0.13$, $p = 0.718$). We now report these results in the revised manuscript (pg. 9).

For example, resolving uncertainty has instrumental utility on agency trials. Thus, to understand developmental change in uncertainty-seeking behavior, it seems important to differentiate the motivation to resolve uncertainty for the sake of resolving uncertainty (e.g., the effect of delta uncertainty on trials in which delta agency was 0 or, especially on trials where the fisherman made the choice for both ponds) and the motivation to resolve uncertainty because it will be helpful in improving one's choices (e.g., the effect of delta uncertainty on trials in which delta agency was > 0.). Along similar lines, the effect of delta EV was interpreted as a motivation to reveal positive information, but it seems important to understand age-related change in revealing positive information when you can then decide whether to choose it vs. positive information about the fisherman's choice. The 'pure' effect of wanting to receive good news across age groups is potentially interesting but not currently reported anywhere (i.e., not just when delta agency is 0, but when both ponds are controlled by the fishermen only).

- Following the reviewer's query, we conducted a new analysis to tease apart the contribution of agency from that of uncertainty and EV. As the reviewer suggest we identified all the trials where $\Delta\text{agency} = 0$ and ran a logistic, mixed effects model on the children's data with information-seeking choices as the dependent variable and ΔEV , $\Delta\text{uncertainty}$, instructions comprehension, EV comparisons score, average correct fishing choices, age and the interaction between age and each other factor as predictors. Once again, we confirmed significant effects of ΔEV (Experiment 1: $\chi^2(1) = 58.09$, $p < 0.001$; Experiment 2: $\chi^2(1) = 85.31$, $p < 0.001$) and $\Delta\text{uncertainty}$ (Experiment 1: $\chi^2(1) = 35.32$, $p < 0.001$; Experiment 2: $\chi^2(1) = 23.59$, $p < 0.001$). As in the main analyses, $\Delta\text{uncertainty}$ positively interacted with age (Experiment 1: $\chi^2(1) = 15.42$, $p < 0.001$; Experiment 2: $\chi^2(1) = 6.06$, $p = 0.014$), while ΔEV did not (Experiment 1: $\chi^2(1) = 2.83$, $p = 0.092$; Experiment 2: $\chi^2(1) = 2.61$, $p = 0.106$). This suggests the effect of uncertainty and age are observed even when agency is constant. We now report this finding in the revised manuscript (pg. 9).
- The reviewer suggested one additional analysis – to test only trials where the fisherman would make the choice on both sides. On average only 3 trials per participant qualified for this subselection which is too small to for a reliable test. However, in Experiment 4 the fisherman always made the decision on both sides for the pond (that is, agency probability was always 0) while $\Delta\text{uncertainty}$ level varied from trial to trial. Thus we were able to examine this in Exp 4. As reported in the Supporting Information, we find significant effects of $\Delta\text{uncertainty}$ here too.
- Together, the results suggest the effects of ΔEV and $\Delta\text{uncertainty}$ on information-seeking reflect a motivation to receive good news and resolve uncertainty, respectively, for their own sake. Moreover, the interaction of these effects with age are observed also when agency is constant across options.

Did the extent to which participants' considered the instrumental value of information actually improve their subsequent fishing choices? This would also be a helpful manipulation check to demonstrate that this information did indeed have instrumental value.

- Yes, considering instrumental value of information improved fishing choices. This is shown by first fitting logistic regression models to each child to obtain their individual coefficients and then correlating the coefficients with proportion of correct fishing choices (i.e. proportion of times the participant chose to fish when the EV of the pond was positive and not to fish when the EV of the pond was negative). We found a significant positive correlation between the extent to which participants considered the instrumental value of information (that is beta coefficient on Δ agency) and the proportion of correct fishing choices (Experiment 1: $r(202) = 0.38$, $p < 0.001$; Experiment 2: $r(193) = 0.27$, $p = 0.001$). We now report this finding in the revised manuscript (pg. 13).

The authors had nice control analyses to convincingly demonstrate that younger children understood the expected value manipulation. But did they understand the uncertainty manipulation? Previous research has found a stronger influence of uncertainty at younger ages, even when multiple motives (e.g., information vs. reward) compete (e.g., Blanco & Sloutsky, 2021). Thus, these results are somewhat in conflict with the literature, and may suggest that they did not compute or track uncertainty in this task.

- The reviewer suggests we test whether the children tracked uncertainty in this task. This is a great suggestion. To that end we ran an additional control experiment ($N=54$). On each trial participants viewed two fishermen sitting on two sides of a river and were asked to guess whether each fisherman was more likely to catch a fish or a can, and to report their confidence about their guess. We used a child friendly scale with images to probe uncertainty. If the children track uncertainty, then the difference in their subjective confidence between the two ponds (Δ confidence) should track the difference in the objective measure of uncertainty we used (that is the difference in the number of seaweed = Δ uncertainty). That is exactly what we found, even in the youngest children. We ran a mixed-effects model predicting Δ confidence from Δ uncertainty, age, instructions comprehension test, and the interaction between age and each of the other factors. Using this model we ran a Johnson-Neyman analysis (which was recommended by Reviewer 3) on all subjects' data. This analysis estimates the effects of Δ uncertainty on Δ confidence at each age. The results revealed that the effect of uncertainty on confidence was already significant at age 4 (estimated β at the lowest tested age = 0.17 ± 0.08 , $p = 0.044$). The same model also revealed the effects of Δ uncertainty were clearly significant across the group as a whole ($\beta = -0.44$, $S.E.M = 0.05$, $\chi^2(1) = 48.37$, $p < 0.001$). We then controlled for average tracking of uncertainty in our original mixed model which predicts information-seeking. We did this by correlating Δ uncertainty and Δ confidence across all trials for each age. We then added these Rs to our mixed-effects model for the main experiment as a covariate (by matching children to the average correlation between

Δ uncertainty and Δ confidence for their corresponding age). Once again we found that information-seeking is predicted by the interaction between Δ uncertainty and age in both Experiment 1 ($\beta = 0.20$, S.E.M = 0.04, $\chi^2(1) = 23.87$, $p < 0.001$) and Experiment 2 ($\beta = 0.14$, S.E.M = 0.04, $\chi^2(1) = 10.31$, $p < 0.001$). This suggests that age-related differences in sensitivity to uncertainty is unlikely to be explained by differences in ability to track uncertainty across age (pg. 13).

Finally, it is somewhat concerning (though perhaps not surprising) that the influence of all variables either remains consistent, or strengthens with age. This seems to suggest that younger participants were making more random choices. It would be helpful to know whether participant reaction times increase or decrease with age. Increases with age would be more concerning as they may just suggest that young children are not really considering much when making their info-seeking decisions.

- Following the reviewer's suggestion, we calculated the correlation between children's age and their average log reaction times across trials. On the contrary, we observe that younger children, on average, tend to take a longer time to respond (Experiment 1: $r(202) = -0.58$, $p < 0.001$; Experiment 2: $r(194) = -0.57$, $p < 0.001$). The results were the same using raw, rather than log, reaction times (Experiment 1: $r(202) = -0.5$, $p < 0.001$; Experiment 2: $r(194) = -0.42$, $p < 0.001$). This is contrary to the possibility that they are simply making random choices. We now report this finding in the revised manuscript (pg. 13).

More minor concerns:

I found the description of the task somewhat difficult to follow. It would be helpful to state participants' goal up front: to make the best 'fishing' choice for 0, 1, or 2 ponds on each trial, which they could do by revealing information about both ponds. Then it would be helpful to state that although participants' goal was to make the best fishing choice, the analyses of interest focused on the information choice at the start of the trial.

- Thank you for this note. We have now rewritten the task description.

Also, were participants incentivized to perform well?

- Participants were informed that their goal is to help the fisherman get the greatest number of fish and avoid cans. They did not receive a bonus for doing so. We now specify this in the revised Methods section (pg. 15).

I also found the giant lists of statistics for many of the analyses very difficult to parse. The tables included in the supplement are much easier to process, and I think most of the results should be reported in these compact tables rather than long lists with each age group and each experiment.

- Following the reviewer's suggestion, we now report tables in the main text in place of the list of statistics as previously reported (pg. 10).

It is not clear why gender was included in the model. Including control variables without sufficient justification can lead to biased estimates (see Wysocki et al., 2022).

- Following the reviewer's suggestion, we removed gender from our control variables.

The methods and results are also reported incorrectly in several places. For both studies, the age range is reported as 10 - 12 and 10-11, and not 4 - 12.

- We thank the reviewer for spotting this error. The age range is now correctly reported as 4-12.

In addition, the description of the models state that delta EV, agency, and uncertainty are included in the model as both fixed and random intercepts. This does not really make any sense, but in looking at the code (thank you for providing that on github!) it seems as though the authors actually included delta EV, agency, and uncertainty as fixed effects and random slopes for each participant, which makes a lot more sense.

- We thank the reviewer for noticing this. We have now corrected our description of the models according to the reviewer's suggestion.

Reviewer #3

In this paper, the authors test children and adults on an information-seeking task in which three key aspects of information (affective, cognitive and instrumental) are experimentally manipulated. In this way, authors can tear apart the specific aspects children focus on when seeking information, and when they start to do so across development. They find that preschoolers are already sensitive to the affective and instrumental components of information, while they become sensitive to the cognitive component (i.e., the level of uncertainty) only from 6 years of age onwards. The second experiment is a direct replication of the first one, and it finds very comparable results, which shows replicability of the findings.

This paper is quite interesting, it focuses on a trending topic tackling it with a novel paradigm, and produces important findings that will help the field move forward. The methodology is very sound, and the additional experiments and explanations about how the variables were manipulated make it even more robust, although it could make some key improvements on the statistical analysis part.

Having said that, I think the paper in its current form makes a minority of claims that are not directly following from the evidence it provides, and this prevents the paper from being accepted in its current form. I address this below. The comments are not in order of importance, but as a heads-up, the two key points concern statistical analyses (the evolution of the effects across time can be investigated in a more convincing manner) and the interpretation (which partly follows from the limits of the current statistical analyses).

- We thank the reviewer for this positive evaluation. As detailed below, we now address all the reviewer’s concerns and follow the reviewer’s helpful suggestions.

Title and line 24: “How children decide”, “how children integrate”. I think this study is more about the “what” than the “how”. It gives us very crucial insights on what children focus on, but it is not mechanistic: it doesn’t get to the cognitive or brain processes that support children’s decisions. For this reason, I am not sure that it directly answers a “how” question.

- Following the reviewer’s comment we have altered the title to *Multifaceted information-seeking motives in children*.

55: To my knowledge, there is no evidence of active information seeking in newborns (for information seeking abilities in infants, see Hunnius, 2022). The evidence cited concerns the second year of life; Please revisit the statement accordingly.

- We thank the reviewer for this observation and for providing a reference. We have now revisited our introductory statement in accordance with the reviewer’s suggestion (pg. 4).

97: “computationally”. This word is used here and a couple of other times across the paper (e.g., line 356). It is not immediately clear to me how the study is computational – it is disentangling different motives experimentally, but I’m not sure it does so computationally. I would either justify the claim or modify it.

- The word ‘computationally’ is used here to explain how we disentangle the weights assigned to the motives. This is meant to imply that the three factors of interest differ and can be quantified on every trial, and by adding them all into one linear regression we can compute the weights assigned to each. We appreciate that different readers may interpret the term differently and thus we delete the word and leave ‘disentangle’ (pg. 5).

98: The authors wrote: “Rather, motives have either been studied in isolation or in a situation where they are confounded”. Please provide references.

- We now add references as requested (pg. 5).

129: Sentence: “Vice versa if they selected the left pond”. Please rephrase.

- This has now been rephrased (pg. 6).

136, 138, and 139: “varied by varying”. Please rephrase.

- This has now been rephrased (pg. 5-6).

156: “the left pond of the pond”: please clarify.

- This should have read “the left side of the pond”. This has been corrected (pg. 6). Thank you for noticing.

164: Children were split in age groups, and groups were treated as a factor, but I think there would be an advantage in keeping age as a continuous variable. I would suggest a slightly

different analysis because the current one also creates problems in the interpretation of the results (for example, it is not possible to conclude that any group has higher or lower beta scores without conducting posthoc tests, see comments below for more information).

Thank to the authors I was able to access the data and the analysis scripts. I introduce the steps I took below, followed by an explanation. I post this script here simply because it is the most efficient way to make my point clear to the authors – this analysis can also be done in other ways, but I do think it comes with many advantages.

- I imported the data with:

```
dat_compt <- read.csv("data_main_experiment_children_1.csv")
```

- Run their script analysis_main_experiment.R

- Run these additional lines at the end of the script (explanation follows):

```
#####
```

```
library(interactions)
```

```
library(modelbased)
```

```
dat_compt2 = dat_compt[c("info_choice",  
"delta_EV","delta_uncertainty_level", "delta_agency", "age_in_months",  
"percent_comprehension","wob", "gender_coded", "subject_ID")]
```

```
dat_compt2 = na.omit(dat_compt2)
```

```
# run the model
```

```
compt_mod <- glmer(info_choice ~ (delta_EV + delta_uncertainty_level +  
delta_agency)*log(age_in_months) +
```

```
percent_comprehension + wob + gender_coded +
```

```
(delta_EV + delta_uncertainty_level + delta_agency
```

```
| subject_ID),
```

```
data = dat_compt2, family = binomial, control = glmerControl(optimizer  
= "bobyqa", optCtrl=list(maxfun=2e5)), nAGQ = 0)
```

```
summary(compt_mod)
```

```
# plot interaction
```

```
interact_plot(compt_mod, pred = delta_agency, modx = age_in_months,  
interval = TRUE, plot.points = FALSE, int.width = 0.8, line.thickness  
= 1, modx.values = c(54, 78, 102, 126, 150 ))
```

```
# estimate predictive effects
```

```
slopes <- estimate_slopes(compt_mod, trend = "delta_agency", at =  
"age_in_months", length = 54)
```

```
# plot significance of the effect across time (Johnson-Neyman plot)
plot(slopes)
# show stats
slopes
#####
```

In the model, time is interacting with the main variables of interest (the deltas). I added a logarithmic function because it seems a sensible idea that these abilities settle at some point across development (and this was also visible from the figures in the paper). When plotting the effects, you can see how they evolve across development. For this reason, I think this kind of analysis is better suited to studying developmental change.

- We are extremely grateful to the reviewer for providing suggested analyses and related code. We now use an adjusted version of the above code in our analyses on children's data (accounting for other reviewer's observations – e.g., removing gender as a covariate and adding correct fishing choices). The new analysis supports our previous conclusions. In particular, the new results confirm that while all three motives (EV, agency, uncertainty) are associated with children's information-seeking choices, the importance of agency (Experiment 1: $\chi^2(1) = 24.19$, $p < 0.001$; Experiment 2: $\chi^2(1) = 10.12$, $p = 0.001$) and uncertainty (Experiment 1: $\chi^2(1) = 23.85$, $p < 0.001$; Experiment 2: $\chi^2(1) = 10.33$, $p = 0.001$) increases as children get older, while the relative importance of EV remains relatively stable from age 4 to 12 (Experiment 1: $\chi^2(1) = 2.86$, $p = 0.091$; Experiment 2: $\chi^2(1) = 0.14$, $p = 0.710$). We define age in years for easier interpretation, but all our results remain consistent regardless of which measure is used to define age. We report the analysis in the revised manuscript (pg. 9-11).
- Note, that when verifying instruction comprehension only we did keep age groups. This is simply to make sure each age group comprehend all aspects of the task.

When plotting the slopes with a Johnson-Neyman plot, you can also identify the precise moment across development when the effect becomes significant.

- Following the reviewer's suggestion we now also show Johnson-Neyman plots in the main text. The analysis revealed that the effect of good news (ΔEV) on information seeking were already significant at age 4 (Experiment 1: $\beta = 0.43 \pm 0.12$, $p < 0.001$; Experiment 2: $\beta = 0.63 \pm 0.14$, $p < 0.001$), as was that of agency (Experiment 1: $\beta = 0.40 \pm 0.13$, $p = 0.002$; Experiment 2: $\beta = 0.40 \pm 0.13$, $p < 0.001$). The tendency to select information when uncertainty was high emerged at around age 5 (Experiment 1: age 5.52, $\beta = 0.11 \pm 0.05$, $p = 0.037$; Experiment 2: age 4.85, $\beta = 0.15 \pm 0.07$, $p = 0.037$). We report the analysis in the revised manuscript (pg. 12).

170: Comprehension scores have been clustered all together, but they concern very different aspects of the task (understanding of the instructions, theory of mind...). Is there any specific

aspect that is affecting the results (vs overall understanding)? I'd like to see the descriptive stats of each question separately – in the response, not necessarily in the paper.

- As requested, we attach the descriptive stats of each comprehension question as a table at the end of this document.

210: The title does not reflect the content of the chapter.

- The reviewer is correct. We now altered this subtitle to *Key independent variables*.

226: What the standard deviation of the items is could be made a bit clearer here, even if it is addressed in methods.

- Following the reviewer's suggestion, we now specify that by standard deviation of the items we mean "the standard deviation from the mean value of all visible items in the pond", i.e., $\sigma = \sqrt{\frac{\sum(x_i - \mu)^2}{N}}$, where x_i is the value of item i (ranging from -20 to 20 as specified in the methods), μ is the mean of all items in the pond, and N is the total number of items in the pond, i.e., six (pg. 9).

249: To my knowledge, logistic regression works with the likelihood ratio chi-square test. Is this the same as the chi-square difference test?

- We thank the reviewer for spotting this typo. We indeed conducted likelihood ratio chi-square tests and have corrected the main text to accurately reflect this (pg. 20).

271: $p = 0.015$ instead of 0.15.

- We thank the reviewer for noticing this typo. Since we no longer do this analysis, the associated sentence is no longer present in the text.

274 "Together, these results suggest that information-seeking choices that help regulate emotion remain stable throughout development." This conclusion of stability is inferred from the lack of an effect. Moreover, if the ability is gradually increasing, testing binned age as independent groups will not pick it up. The analysis I suggest above would also help with this: If the beta coefficients for interaction between time and main independent variables are significantly different from zero, there is developmental change. If they are not different from zero, it is still unwarranted to make conclusions about stability, unless Bayesian stats and Bayes factors for the null hypotheses are reported (or at least effect sizes for frequentist tests).

- Following the reviewers' comment we used Bayesian statistic to examine if the null relationship between ΔEV and age (years, continuous) is reliable. Indeed, this was the case (Experiment 1: $BF = 0.36$; Experiment 2: $BF = 0.17$). We now report this in the revised manuscript (pg. 10).

280: "which becomes even greater". How has this been tested? Similar to the previous point for line 274.

- As this reviewer recommends, we no longer bin children into groups in our analysis, thus this sentence has now been deleted.

291 and 307: You may need posthoc tests or analysis as described in my comment for line 164 if you want to better characterize this increase.

- Yes, we now use the analysis the reviewer recommended regarding line 164 (pg. 12).

498: the correlations are reported and I was convinced that the independent variables are not correlated with each other – which is a really good aspect of the design. As a suggestion – It would be more convincing to have a figure with an example sequence of trials, with trial number on the x-axis and values of the three main dependent variables on the y-axis.

- We now add an example sequence of trials exactly as suggested by the reviewer (pg. 1 of the Supporting Information).

Final comment:

The theoretical framework is robust, supported by evidence, and explains the data well. However, in the current formulation of the paper, it might come across that the framework is also complete, in the sense that all kinds of information seeking can be explained through it. However, some drives for information are partly unrelated, especially coming from the developmental literature, where infants' information seeking has traditionally been related to novelty/complexity (see Kidd & Hayden, 2015 for a review) and more recently to learning progress (see Poli, Serino, Mars & Hunnius, 2020). Learning progress has also been related to adults' exploratory behaviour in very recent studies (Poli, Meyer, Mars, & Hunnius, 2022; Ten, Kaushik, Oudeyer, & Gottlieb, 2021). These drives cannot be addressed within the current study; For example, there is no opportunity to learn or to get better at the task, which makes it difficult to study learning progress; the elements never change, which makes it difficult to study novelty. I would either explain how that literature relates to the current theory (are all of these different aspects of the cognitive part of this theory, maybe?), or at least acknowledge that more drives than the ones addressed here exist.

- Thank you for this important suggestion. We believe these additional factors (such as learning and novelty) may be part of the motives we identify in our theoretical paper (Sunstein & Sharot 2020) – namely 'instrumental utility' and 'cognitive utility'. In our theory 'cognitive utility' relates to the ability of information to increase understanding of the world in which the agent operates (even when information is non-instrumental). 'Instrumental utility' is the ability of information to guide choices that will lead to external rewards. Learning progress and novelty are related to cognitive utility and often (but not always) to instrumental utility. We now explain how these factors relate to our theoretical framework (pg. 14). Here, we test specific aspects of cognitive utility and instrumental utility (namely uncertainty reduction and agency) and it would be very interesting to test the additional factors the reviewer highlights. We add a discussion of these issues in the manuscript (pg. 14).

Related to this point, the current theory succeeds in capturing the multifaceted nature of information seeking. The breadth of its aim is excellent, but future research will need to move towards greater depth as well. For example, once we've acknowledged that information is driven by cognitive aspects, we need to determine how exactly this happens, and what are the mechanisms that underlie this information seeking: for example, an attunement to learning progress might be the policy that allows agents to maximize information (see Oudeyer's research). This is not a limitation of the current study – The study is robust and it delivers what it promised (which is a lot, in my opinion). Rather, this is a point for future research that I believe it's worth mentioning.

- We agree. These avenues are fascinating and exciting, and we now mention them in the discussion (pg. 14).

Once again, we would like to thank you for allowing us to make adjustments to our manuscript according to the helpful comments of the reviewers. The insights, specific suggestions and code provided by the reviewers were truly exceptional and strengthened the paper considerably, which we hope is now deemed suitable for *Nature Communications*.

Best Regards,

Gaia Molinaro and Tali Sharot,

University of California at Berkeley, UCL & MIT

Exp.	Q. #	Question	Age	Mean	SD
1	1	Which one is better? Big fish or big can?	4-5	2.00	0.00
1	2	Which one is better? Big fish or small fish?	4-5	1.62	0.57
1	3	Which one is better? Small fish or small can?	4-5	1.88	0.33
1	4	Which one is worse? Big can or small can?	4-5	1.92	0.27
1	5	Who will decide whether to fish or not when you see this? You, the Fishertwin, or we don't know yet?	4-5	1.81	0.57
1	6	Who will decide whether to fish or not when you see this? You, the Fishertwin, or we don't know yet?	4-5	1.77	0.43
1	7	Who will decide whether to fish or not when you see this? You, the Fishertwin, or we don't know yet?	4-5	1.54	0.71
1	8	What happens if you click on the flashlight button? No objects will be shown, only the non-greasy objects will be shown, or nothing happens?	4-5	1.46	0.76
1	9	What happens if you click this button? You will fish and the Fishertwin will put your catch in the basket, or you will not fish?	4-5	1.96	0.20
1	10	What happens if you click this button? You will fish and the Fishertwin will put your catch in the basket, or you will not fish?	4-5	1.85	0.37
1	11	Can the Fishertwin see what happens when you use the flashlight? Yes or no?	4-5	1.69	0.47
1	12	Is there something behind the seaweed? Yes, no, or maybe?	4-5	1.88	0.33
1	13	What's behind the seaweed? A fish, a can, or we don't know?	4-5	1.73	0.53
1	1	Which one is better? Big fish or big can?	6-7	2.00	0.00
1	2	Which one is better? Big fish or small fish?	6-7	1.96	0.19
1	3	Which one is better? Small fish or small can?	6-7	1.96	0.19
1	4	Which one is worse? Big can or small can?	6-7	1.89	0.32
1	5	Who will decide whether to fish or not when you see this? You, the Fishertwin, or we don't know yet?	6-7	1.70	0.54
1	6	Who will decide whether to fish or not when you see this? You, the Fishertwin, or we don't know yet?	6-7	1.96	0.19
1	7	Who will decide whether to fish or not when you see this? You, the Fishertwin, or we don't know yet?	6-7	1.78	0.51

1	8	What happens if you click on the flashlight button? No objects will be shown, only the non-greasy objects will be shown, or nothing happens?	6-7	1.85	0.53
1	9	What happens if you click this button? You will fish and the Fishertwin will put your catch in the basket, or you will not fish?	6-7	2.00	0.00
1	10	What happens if you click this button? You will fish and the Fishertwin will put your catch in the basket, or you will not fish?	6-7	1.96	0.19
1	11	Can the Fishertwin see what happens when you use the flashlight? Yes or no?	6-7	1.85	0.36
1	12	Is there something behind the seaweed? Yes, no, or maybe?	6-7	1.89	0.32
1	13	What's behind the seaweed? A fish, a can, or we don't know?	6-7	1.81	0.48
1	1	Which one is better? Big fish or big can?	8-9	2.00	0.00
1	2	Which one is better? Big fish or small fish?	8-9	2.00	0.00
1	3	Which one is better? Small fish or small can?	8-9	2.00	0.00
1	4	Which one is worse? Big can or small can?	8-9	1.92	0.28
1	5	Who will decide whether to fish or not when you see this? You, the Fishertwin, or we don't know yet?	8-9	2.00	0.00
1	6	Who will decide whether to fish or not when you see this? You, the Fishertwin, or we don't know yet?	8-9	2.00	0.00
1	7	Who will decide whether to fish or not when you see this? You, the Fishertwin, or we don't know yet?	8-9	1.96	0.20
1	8	What happens if you click on the flashlight button? No objects will be shown, only the non-greasy objects will be shown, or nothing happens?	8-9	2.00	0.00
1	9	What happens if you click this button? You will fish and the Fishertwin will put your catch in the basket, or you will not fish?	8-9	2.00	0.00
1	10	What happens if you click this button? You will fish and the Fishertwin will put your catch in the basket, or you will not fish?	8-9	2.00	0.00
1	11	Can the Fishertwin see what happens when you use the flashlight? Yes or no?	8-9	1.92	0.28
1	12	Is there something behind the seaweed? Yes, no, or maybe?	8-9	1.96	0.20
1	13	What's behind the seaweed? A fish, a can, or we don't know?	8-9	2.00	0.00

1	1	Which one is better? Big fish or big can?	10-12	2.00	0.00
1	2	Which one is better? Big fish or small fish?	10-12	2.00	0.00
1	3	Which one is better? Small fish or small can?	10-12	2.00	0.00
1	4	Which one is worse? Big can or small can?	10-12	1.96	0.20
1	5	Who will decide whether to fish or not when you see this? You, the Fishertwin, or we don't know yet?	10-12	1.96	0.20
1	6	Who will decide whether to fish or not when you see this? You, the Fishertwin, or we don't know yet?	10-12	2.00	0.00
1	7	Who will decide whether to fish or not when you see this? You, the Fishertwin, or we don't know yet?	10-12	1.88	0.43
1	8	What happens if you click on the flashlight button? No objects will be shown, only the non-greasy objects will be shown, or nothing happens?	10-12	2.00	0.00
1	9	What happens if you click this button? You will fish and the Fishertwin will put your catch in the basket, or you will not fish?	10-12	2.00	0.00
1	10	What happens if you click this button? You will fish and the Fishertwin will put your catch in the basket, or you will not fish?	10-12	2.00	0.00
1	11	Can the Fishertwin see what happens when you use the flashlight? Yes or no?	10-12	2.00	0.00
1	12	Is there something behind the seaweed? Yes, no, or maybe?	10-12	1.88	0.33
1	13	What's behind the seaweed? A fish, a can, or we don't know?	10-12	1.96	0.20
2	1	Which one is better? Big fish or big can?	4-5	1.98	0.15
2	2	Which one is better? Big fish or small fish?	4-5	1.67	0.48
2	3	Which one is better? Small fish or small can?	4-5	1.93	0.26
2	4	Which one is worse? Big can or small can?	4-5	1.83	0.38
2	5	Who will decide whether to fish or not when you see this? You, the Fishertwin, or we don't know yet?	4-5	1.71	0.51
2	6	Who will decide whether to fish or not when you see this? You, the Fishertwin, or we don't know yet?	4-5	1.76	0.43
2	7	Who will decide whether to fish or not when you see this? You, the Fishertwin, or we don't know yet?	4-5	1.79	0.42
2	8	What happens if you click on the flashlight button? No objects will be shown, only the non-greasy objects will be shown, or nothing happens?	4-5	1.76	0.48

2	9	What happens if you click this button? You will fish and the Fishertwin will put your catch in the basket, or you will not fish?	4-5	1.98	0.15
2	10	What happens if you click this button? You will fish and the Fishertwin will put your catch in the basket, or you will not fish?	4-5	1.95	0.22
2	11	Can the Fishertwin see what happens when you use the flashlight? Yes or no?	4-5	1.69	0.47
2	12	Is there something behind the seaweed? Yes, no, or maybe?	4-5	1.93	0.26
2	13	What's behind the seaweed? A fish, a can, or we don't know?	4-5	1.79	0.47
2	2	Which one is better? Big fish or small fish?	6-7	1.98	0.14
2	3	Which one is better? Small fish or small can?	6-7	1.90	0.36
2	4	Which one is worse? Big can or small can?	6-7	1.88	0.32
2	5	Who will decide whether to fish or not when you see this? You, the Fishertwin, or we don't know yet?	6-7	1.83	0.38
2	6	Who will decide whether to fish or not when you see this? You, the Fishertwin, or we don't know yet?	6-7	1.87	0.34
2	7	Who will decide whether to fish or not when you see this? You, the Fishertwin, or we don't know yet?	6-7	1.73	0.53
2	8	What happens if you click on the flashlight button? No objects will be shown, only the non-greasy objects will be shown, or nothing happens?	6-7	1.92	0.27
2	9	What happens if you click this button? You will fish and the Fishertwin will put your catch in the basket, or you will not fish?	6-7	1.96	0.19
2	10	What happens if you click this button? You will fish and the Fishertwin will put your catch in the basket, or you will not fish?	6-7	1.98	0.14
2	11	Can the Fishertwin see what happens when you use the flashlight? Yes or no?	6-7	1.75	0.44
2	12	Is there something behind the seaweed? Yes, no, or maybe?	6-7	1.83	0.43
2	13	What's behind the seaweed? A fish, a can, or we don't know?	6-7	1.73	0.45
2	1	Which one is better? Big fish or big can?	8-9	1.98	0.14
2	2	Which one is better? Big fish or small fish?	8-9	1.98	0.14
2	3	Which one is better? Small fish or small can?	8-9	2.00	0.00

2	4	Which one is worse? Big can or small can?	8-9	1.98	0.14
2	5	Who will decide whether to fish or not when you see this? You, the Fishertwin, or we don't know yet?	8-9	1.90	0.31
2	6	Who will decide whether to fish or not when you see this? You, the Fishertwin, or we don't know yet?	8-9	1.94	0.24
2	7	Who will decide whether to fish or not when you see this? You, the Fishertwin, or we don't know yet?	8-9	1.88	0.39
2	8	What happens if you click on the flashlight button? No objects will be shown, only the non-greasy objects will be shown, or nothing happens?	8-9	2.00	0.00
2	9	What happens if you click this button? You will fish and the Fishertwin will put your catch in the basket, or you will not fish?	8-9	2.00	0.00
2	10	What happens if you click this button? You will fish and the Fishertwin will put your catch in the basket, or you will not fish?	8-9	2.00	0.00
2	11	Can the Fishertwin see what happens when you use the flashlight? Yes or no?	8-9	1.94	0.24
2	12	Is there something behind the seaweed? Yes, no, or maybe?	8-9	1.83	0.38
2	13	What's behind the seaweed? A fish, a can, or we don't know?	8-9	1.96	0.20
2	1	Which one is better? Big fish or big can?	10-12	1.98	0.14
2	2	Which one is better? Big fish or small fish?	10-12	1.96	0.19
2	3	Which one is better? Small fish or small can?	10-12	1.98	0.14
2	4	Which one is worse? Big can or small can?	10-12	1.93	0.26
2	5	Who will decide whether to fish or not when you see this? You, the Fishertwin, or we don't know yet?	10-12	1.94	0.23
2	6	Who will decide whether to fish or not when you see this? You, the Fishertwin, or we don't know yet?	10-12	2.00	0.00
2	7	Who will decide whether to fish or not when you see this? You, the Fishertwin, or we don't know yet?	10-12	1.94	0.23
2	8	What happens if you click on the flashlight button? No objects will be shown, only the non-greasy objects will be shown, or nothing happens?	10-12	2.00	0.00
2	9	What happens if you click this button? You will fish and the Fishertwin will put your catch in the basket, or you will not fish?	10-12	2.00	0.00

2	10	What happens if you click this button? You will fish and the Fishertwin will put your catch in the basket, or you will not fish?	10-12	2.00	0.00
2	11	Can the Fishertwin see what happens when you use the flashlight? Yes or no?	10-12	1.91	0.29
2	12	Is there something behind the seaweed? Yes, no, or maybe?	10-12	1.98	0.14
2	13	What's behind the seaweed? A fish, a can, or we don't know?	10-12	1.91	0.29
6	1	Which one is better? Big fish or big can?	Adults	2	0
6	2	Which one is better? Big fish or small fish?	Adults	2	0
6	3	Which one is better? Small fish or small can?	Adults	1.96	0.19
6	4	Which one is worse? Big can or small can?	Adults	1.89	0.31
6	5	Who will decide whether to fish or not when you see this? You, the Fishertwin, or we don't know yet?	Adults	2	0
6	6	Who will decide whether to fish or not when you see this? You, the Fishertwin, or we don't know yet?	Adults	2	0
6	7	Who will decide whether to fish or not when you see this? You, the Fishertwin, or we don't know yet?	Adults	1.93	0.26
6	8	What happens if you click on the flashlight button? No objects will be shown, only the non-greasy objects will be shown, or nothing happens?	Adults	2	0
6	9	What happens if you click this button? You will fish and the Fishertwin will put your catch in the basket, or you will not fish?	Adults	2	0
6	10	What happens if you click this button? You will fish and the Fishertwin will put your catch in the basket, or you will not fish?	Adults	2	0
6	11	Can the Fishertwin see what happens when you use the flashlight? Yes or no?	Adults	1.96	0.19
6	12	Is there something behind the seaweed? Yes, no, or maybe?	Adults	1.96	0.19
6	13	What's behind the seaweed? A fish, a can, or we don't know?	Adults	2	0
7	1	Which one is better? Big fish or big can?	Adults	2.00	0.00
7	2	Which one is better? Big fish or small fish?	Adults	2.00	0.00
7	3	Which one is better? Small fish or small can?	Adults	2.00	0.00
7	4	Which one is worse? Big can or small can?	Adults	1.93	0.26

7	5	Who will decide whether to fish or not when you see this? You, the Fishertwin, or we don't know yet?	Adults	2.00	0.00
7	6	Who will decide whether to fish or not when you see this? You, the Fishertwin, or we don't know yet?	Adults	2.00	0.00
7	7	Who will decide whether to fish or not when you see this? You, the Fishertwin, or we don't know yet?	Adults	1.93	0.26
7	8	What happens if you click on the flashlight button? No objects will be shown, only the non-greasy objects will be shown, or nothing happens?	Adults	2.00	0.00
7	9	What happens if you click this button? You will fish and the Fishertwin will put your catch in the basket, or you will not fish?	Adults	2.00	0.00
7	10	What happens if you click this button? You will fish and the Fishertwin will put your catch in the basket, or you will not fish?	Adults	2.00	0.00
7	11	Can the Fishertwin see what happens when you use the flashlight? Yes or no?	Adults	1.93	0.26
7	12	Is there something behind the seaweed? Yes, no, or maybe?	Adults	1.93	0.26
7	13	What's behind the seaweed? A fish, a can, or we don't know?	Adults	2.00	0.00

Reviewers' Comments:

Reviewer #1:

Remarks to the Author:

This is a great paper, I am very happy with all the changes and additional analyses that the authors did. Looking forward seeing this paper in print!

Reviewer #2:

Remarks to the Author:

I think the authors have done a very nice job addressing my concerns, and those of the other reviewers. I think the additional models with the interactive effects are particularly interesting (though somewhat surprising), and the control experiment examining uncertainty is also very useful for the interpretation of the results. I have a few minor comments remaining.

1. The authors should be a bit more cautious when making claims about the age at which the influence of different factors on information seeking "emerge" as the exact developmental timepoint at which an effect becomes significant is likely highly specific to the task used.
2. The analysis of RT data is helpful in ensuring that children are not just responding without deliberation, but the fact that all variables show increasing influence with age does indeed suggest that children are acting more 'randomly' (in the sense that their data is less well-captured by the statistical model). So I think the authors need to just adjust their interpretation of these RT results accordingly.
3. The increase in uncertainty-seeking across development is in contrast to many other developmental studies (e.g., Blanco & Sloutsky, 2020; Schulz et al., 2019). It would be helpful for the authors to mention this divergence in their discussion and perhaps speculate about why they see different effects here.

Reviewer #3:

Remarks to the Author:

I was pleased to see that all my comments were addressed in great detail. The analysis section, which I was more concerned about, has changed and improved substantially. The new analyses show how the effects of interest change across development, making the paper's claims more convincing.

Regarding the discussion, I was happy to see that these findings have been linked to the rest of the developmental literature on novelty-, learning-, and information-seeking. I only have one minor comment left, as the authors cite Poli et al. (2020) *Science Advances*, in the paper (p.14, line 402), but Poli et al. (2022) *Cognition*, in the references. The two references are complementary in addressing learning progress and information gain theories (full references below). More broadly, please make sure references are accurate before the final submission.

I now consider the paper ready for publication.

Best wishes,
Francesco

References

Poli, F., Serino, G., Mars, R. B., & Hunnius, S. (2020). Infants tailor their attention to maximize learning. *Science advances*, 6(39), eabb5053.

Poli, F., Meyer, M., Mars, R. B., & Hunnius, S. (2022). Contributions of expected learning progress

and perceptual novelty to curiosity-driven exploration. *Cognition*, 225, 105119.

We would like to thank the reviewers and the editorial team for their continued work and support. To ensure we address all final requests, and for ease of reference, we include the reviews and suggestions below (in bold) followed by our response to each.

Reviewer #1

This is a great paper, I am very happy with all the changes and additional analyses that the authors did. Looking forward seeing this paper in print!

- We thank the reviewer for the positive feedback!

Reviewer #2

I think the authors have done a very nice job addressing my concerns, and those of the other reviewers. I think the additional models with the interactive effects are particularly interesting (though somewhat surprising), and the control experiment examining uncertainty is also very useful for the interpretation of the results. I have a few minor comments remaining.

- We thank the reviewer for the positive feedback and address the remaining minor comments below.

1. The authors should be a bit more cautious when making claims about the age at which the influence of different factors on information seeking “emerge” as the exact developmental timepoint at which an effect becomes significant is likely highly specific to the task used.

- We thank the reviewer for their suggestion. We now state in the discussion that the exact age in which the influence emerges can be task specific (p. 17).

2. The analysis of RT data is helpful in ensuring that children are not just responding without deliberation, but the fact that all variables show increasing influence with age does indeed suggest that children are acting more ‘randomly’ (in the sense that their data is less well-captured by the statistical model). So I think the authors need to just adjust their interpretation of these RT results accordingly.

- Thank you for your comment. We no longer suggest that the RT data can be understood as an indication of less/more randomness (p. 13).

3. The increase in uncertainty-seeking across development is in contrast to many other developmental studies (e.g., Blanco & Sloutsky, 2020; Schulz et al., 2019). It would be helpful for the authors to mention this divergence in their discussion and perhaps speculate about why they see different effects here.

- We thank the reviewer for this additional opportunity to integrate other research into our discussion. As was previously mentioned in our manuscript, “The findings do not suggest that uncertainty reduction is not important in driving information-seeking at this age, but rather that in complex situations where multiple information-seeking motives compete, the

relative importance of uncertainty in guiding information-seeking is low (for a related discussion see Sehl et al., 2021).” We now include specific mention of the studies cited by the reviewer and clarify our point by adding the following sentence: “This may explain the fact that studies with simpler designs have observed uncertainty-guided information-seeking even in children younger than 5 years old (e.g., Blanco & Sloutsky; Schultz et al., 2019).” (p. 17).

Reviewer #3

I was pleased to see that all my comments were addressed in great detail. The analysis section, which I was more concerned about, has changed and improved substantially. The new analyses show how the effects of interest change across development, making the paper's claims more convincing.

- We thank the reviewer for the positive feedback.

Regarding the discussion, I was happy to see that these findings have been linked to the rest of the developmental literature on novelty-, learning-, and information-seeking. I only have one minor comment left, as the authors cite Poli et al. (2020) *Science Advances*, in the paper (p.14, line 402), but Poli et al. (2022) *Cognition*, in the references. The two references are complementary in addressing learning progress and information gain theories (full references below). More broadly, please make sure references are accurate before the final submission.

- Thank you for noticing this typo. Both articles are now referenced in text (p. 18) and reported in reference list (pp. 27, 31).

I now consider the paper ready for publication.

Best wishes,

Francesco

Once again, we would like to thank the reviewers for contributing to our manuscript and we hope it is now deemed suitable for publication in *Nature Communications*.

Best Regards,

Gaia Molinaro and Tali Sharot (University of California at Berkeley, UCL & MIT)